# Stimulation strength controls the rate of initiation but not the molecular organisation of TCR-induced signalling

Claire Y Ma[1], John C Marioni[2,3,4†]*, Gillian M Griffiths[1†]*, Arianne C Richard[1,2†]*

[1]Cambridge Institute for Medical Research, University of Cambridge, Cambridge, United Kingdom; [2]Cancer Research UK Cambridge Institute, University of Cambridge, Cambridge, United Kingdom; [3]EMBL-European Bioinformatics Institute, Wellcome Genome Campus, Cambridge, United Kingdom; [4]Wellcome Sanger Institute, Wellcome Genome Campus, Cambridge, United Kingdom

**Abstract** Millions of naïve T cells with different TCRs may interact with a peptide-MHC ligand, but very few will activate. Remarkably, this fine control is orchestrated using a limited set of intracellular machinery. It remains unclear whether changes in stimulation strength alter the programme of signalling events leading to T cell activation. Using mass cytometry to simultaneously measure multiple signalling pathways during activation of murine CD8$^+$ T cells, we found a programme of distal signalling events that is shared, regardless of the strength of TCR stimulation. Moreover, the relationship between transcription of early response genes *Nr4a1* and *Irf8* and activation of the ribosomal protein S6 is also conserved across stimuli. Instead, we found that stimulation strength dictates the rate with which cells initiate signalling through this network. These data suggest that TCR-induced signalling results in a coordinated activation program, modulated in rate but not organization by stimulation strength.

*For correspondence:
marioni@ebi.ac.uk (JCM);
gg305@cam.ac.uk (GMG);
acr62@cam.ac.uk (ACR)

†These authors contributed equally to this work

Competing interests: The authors declare that no competing interests exist.

## Introduction

Effector differentiation of a naïve CD8$^+$ T cell begins when its T cell receptor (TCR) recognizes a peptide-MHCI ligand complex. If the interaction is strong enough, a cascade of signalling events follows that allows the naïve T cell to differentiate and expand into a pool of effector cells. Signal transduction downstream of the TCR involves a highly diverse network of post-translational protein modifications that ultimately drive transcriptional, translational, metabolic and cytoskeletal changes in the cell. It is estimated that fewer than 0.01% of naïve CD8$^+$ T cells can recognize a particular foreign peptide-MHCI complex (*Jenkins and Moon, 2012*). Despite the diversity of rearranged TCRs on these naïve cells and the extensive range of antigenic peptides that may be presented, the intracellular machinery within each naïve T cell is able to sense the strength of the receptor-ligand interaction and mount an appropriate response.

Previous work has demonstrated that the strength of stimulation a T cell receives upon binding a peptide-MHC ligand complex determines its fate in the thymus and its probability of activating in the periphery. During thymic selection, T cells that weakly recognize self-peptides are retained, while those that strongly recognize self-peptides undergo negative selection and are removed (*Daniels et al., 2006*; *Hogquist et al., 1994*; *Juang et al., 2010*; *Prasad et al., 2009*). In the periphery, the population response to stimuli of different strengths can vary in speed, magnitude and phenotype (*Denton et al., 2011*; *King et al., 2012*; *Moreau et al., 2012*; *Ozga et al., 2016*; *Palmer et al., 2016*; *Zehn et al., 2009*). Work from our group and others indicates that these observations may be explained by the fact that stimulation strength controls the rate with which individual cells activate transcriptional and proliferative processes (*Hommel and Hodgkin, 2007*;

**eLife digest** Amongst the different types of cells the body uses to protect itself, killer T cells have an unique role: they can detect and neutralize cells that have been become dangerous for the organism – for example, cells which are cancerous or hijacked by viruses.

In a healthy organism, T cells circulate through the body in an inactivated state. When a disease emerges, receptors at the surface of the cells can detect elements coming from harmful agents; this stimulation then triggers a molecular cascade inside the T cell that leads to activation. This system is relatively simple, pairing a finite number of receptors with a limited set of internal components.

At the same time, the activity of T cells is finely regulated, and their activation tightly controlled: they must kill enough cells to stop the illness without causing excess damage. How this is accomplished is still unclear. A T cell can recognize harmful agents that bind its receptors with differing strengths, but how this variability in stimulation strength affects the signaling processes within the cell is still poorly understood.

To investigate this question, Ma et al. used an approach called mass cytometry and analyzed the internal processes of mouse killer T cells receiving different strengths of stimulation. This investigation revealed little change in the patterns of signaling in response to signals of different strength. Instead, what differed was the proportion of T cells that became activated, and how fast this process took place: stronger stimulations led to a larger population of killer T cells being activated more rapidly. Overall, this work sheds light on how killer T cells fine-tune their response to illness using only a simple system to control their activation.

*Richard et al., 2018*). This then raises the question, how does stimulation strength impact signalling downstream of the TCR, and how does this relate to transcriptional activation (*Balyan et al., 2018*)?

Many previous studies have examined signalling mediators and their coordinated network during naïve T cell activation (*Kannan et al., 2012*; *Krishnaswamy et al., 2014*; *Voisinne et al., 2019*). Signalling through the TCR (*Courtney et al., 2018*; *Navarro and Cantrell, 2014*) begins with LCK and Fyn-mediated tyrosine phosphorylation of ITAM motifs on the invariant CD3 subunits. This creates a high affinity binding site for ZAP70, which, upon phosphorylation and activation, leads to the generation of the LAT-SLP76 signalosome. From here, signalling activates multiple cascades including MAPK (MEK1/2-ERK1/2), PDK1-PI3K, calcium, and NFκB (including IκBα-p65) pathways, each amplified and propagated via a series of phosphorylation events or other post-translational modifications. Signal transduction pathways can be broadly categorized as digital, with distinct 'on' or 'off' outcomes, or analogue, giving rise to a graded response (*Conley et al., 2016*; *Zikherman and Au-Yeung, 2015*). In digital signalling, once the threshold for activation is surpassed, an output signal of constant intensity is produced. In analogue signalling, higher intensity of the originating stimulus results in a proportionally higher intensity of the output signal.

Previous work has demonstrated that ligand potency determines the extent of signalling at various proximal and distal nodes (*Palmer et al., 2016*; *Rosette et al., 2001*). Studies focused on digital signalling nodes showed that stimulation strength affects the percentage of cells that phosphorylate ERK (*Altan-Bonnet and Germain, 2005*; *Das et al., 2009*; *Tian et al., 2007*), PKD2 (*Navarro et al., 2014*), IκBα and the p65 component of NFκB (*Kingeter et al., 2010*). Most of these studies have looked at each signalling molecule separately. It therefore remains unclear how ligand potency affects the coordination of signalling downstream of the TCR in naïve T cells.

TCR-induced responses are rapid and often transient, and responding cell populations can be heterogeneous. Single-cell approaches are therefore well-suited to examining this system. Mass cytometry provides single-cell resolution, uses antibody-mediated measurements that can detect post-translational signalling protein modifications, and can achieve high-dimensionality through simultaneous measurement of up to 40 epitopes (*Bandura et al., 2009*; *Bendall et al., 2011*; *Lou et al., 2007*; *Ornatsky et al., 2010*). Previous mass cytometry studies of T cell signalling have demonstrated that small differences in proximal signalling molecules are propagated and amplified in downstream targets (*Mingueneau et al., 2014*) and that the interplay of 'activatory' versus 'inhibitory' signalling components determines the response of effector T cells to different antigen doses (*Wolchinsky et al., 2014*).

In this study, we designed a mass cytometry panel probing surface receptors and elements of key signalling pathways (*Figure 1*) to examine the effect of stimulation strength on naïve CD8[+] T cell responses. We used a minimal antigen presentation system to ask how modulating only the strength of the TCR-pMHC interaction affects signalling pathways without the influence of variable costimulatory factors or feedback from other cell types. Our multi-dimensional approach allowed us to determine how ligand potency impacts the synchronization of multiple parallel pathways. Through simultaneous measurement of S6 phosphorylation and early mRNA transcripts, we also examined the concurrent activation of these markers of translational and transcriptional processes. Our data suggest that the coordination of the TCR-induced signalling pathways that we tested does not vary with stimulation strength. Instead, strength of stimulation determines the rate with which T cells initiate this programme.

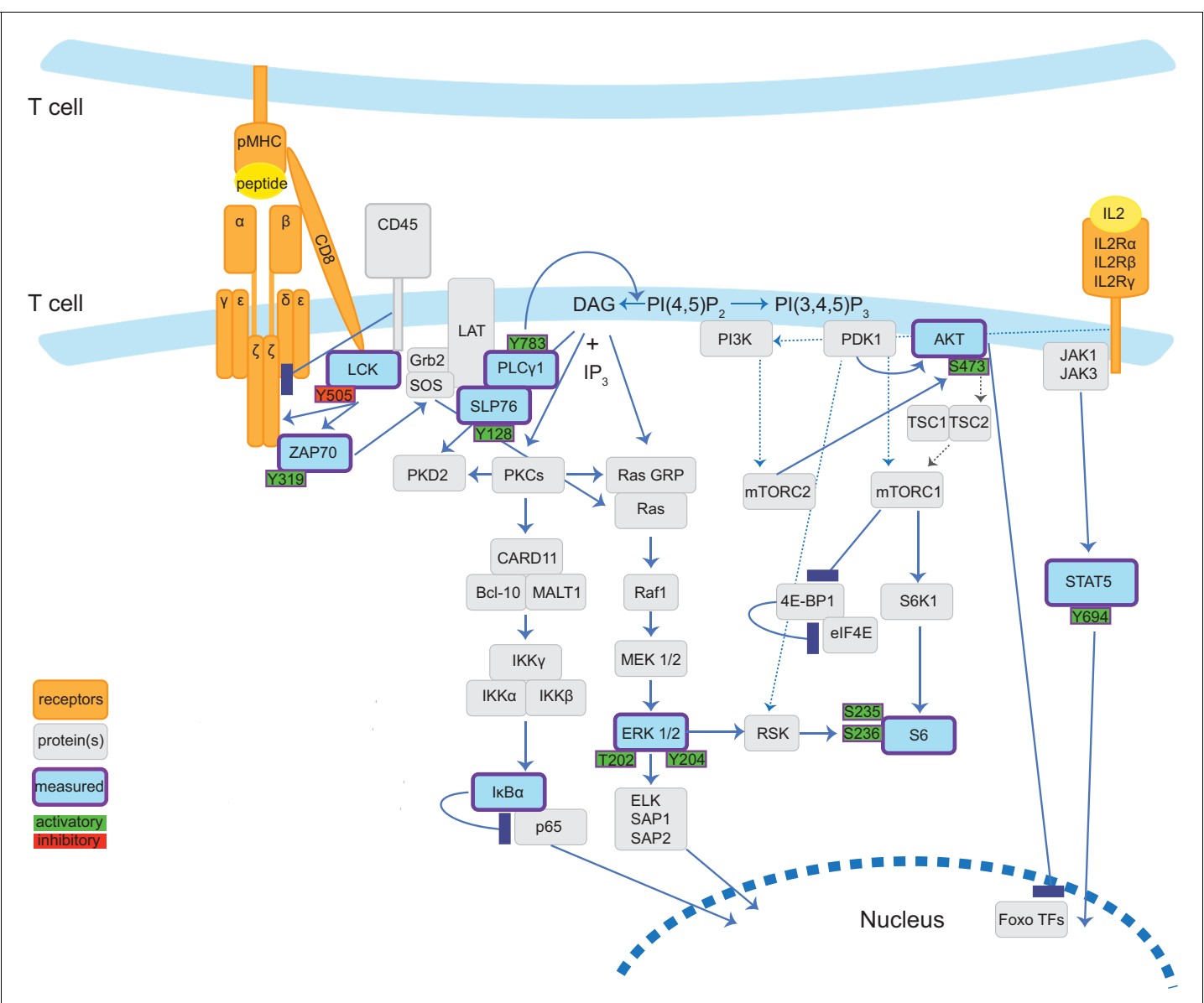

**Figure 1.** Diagrammatic representation of TCR signalling pathways measured by mass cytometry panel. Cartoon depicts the TCR-related signalling pathways examined in this study in our minimal stimulation system wherein T cells present antigen to each other. Solid lines indicate evidence of direct and dotted lines suggested or indirect interaction. Signalling proteins and post-translational modifications directly measured by mass cytometry antibodies are coloured blue (proteins) or green/red (sites of phosphorylation events) and outlined in purple. The mass cytometry panel also profiled surface proteins TCRβ, CD8α, CD45, CD25 (IL2Rα), and CD44.

## Results

### Mass cytometry detects active T cell conjugates

We used the OT-I TCR transgenic mouse on a recombination-activating gene (RAG)-deficient background as a model for evaluating the impact of stimulation strength on TCR signalling pathways. In this model, all peripheral CD8[+] T cells recognize the ovalbumin peptide SIINFEKL. Variants of SIINFEKL with altered potency for the OT-I TCR allow manipulation of the strength of TCR stimulation (*Daniels et al., 2006*; *Hogquist et al., 1994*; *Hong et al., 2018*; *Rosette et al., 2001*). In this study, we used the high potency SIINFEKL (N4), intermediate potency SIITFEKL (T4), and low potency SIIGFEKL (G4) peptides, as well as an unrelated control peptide, ASNENMDAM (NP68).

We designed a custom mass cytometry antibody panel to detect five surface markers of T cell identity and activation, eight phosphorylated signalling proteins with corresponding total proteins, and IκBα, which is degraded in response to stimulation (*Figure 1*; Materials and methods). The antibodies labelled key components of major signalling pathways, including proximal signalling (pZAP70 [Y319]/pSyk[Y352], pSLP76[Y128], pLCK[Y505] and pPLCγ1[Y783]), the MAPK pathway (pERK1/2 [T202/Y204]), the PDK1-PI3K and mTOR pathways (pAKT[S473], pS6[S235/S236]), the NFκB pathway (IκBα) and the IL2 pathway (pSTAT5[Y694]). All of these phosphorylation sites indicate active signalling with the exception of the inhibitory Y505 phosphorylation of LCK (*D'Oro and Ashwell, 1999*; *Marth et al., 1988*). Measurement of total proteins allowed us to determine whether changes in phospho-protein staining were due to differences in signalling or protein expression levels.

We isolated naïve CD8[+] T cells from OT-I *Rag1*[-/-] splenocytes before stimulating with ligands of various strengths. To monitor signalling while naïve CD8[+] T cells transitioned to activated T cells, and to relate signalling to changes in mRNA and protein expression during this process, cells were stimulated for 1, 2, 4 and 6 hr. We used a minimal, controlled system of peptide addition, allowing T cells to present antigens to each other. We also added exogenous IL2 to mitigate effects of potency-dependent differences in paracrine IL2 (*Au-Yeung et al., 2017*; *Denton et al., 2011*; *Marchingo et al., 2014*; *Tan et al., 2017*; *Verdeil et al., 2006*; *Voisinne et al., 2015*) and provide all cells with an effector-promoting environment (*Pipkin et al., 2010*; *Verdeil et al., 2006*). This system was chosen in order to examine the cell-intrinsic effects of TCR stimulation strength on signalling pathways. Peptides were added at 1 μM since peptide titration revealed minimal differences in the percentage of cells phosphorylating S6 and ERK between 100 nM and 1 μM stimulation conditions (*Figure 2—figure supplement 1*). Using this reductionist stimulation system, we previously found that stimulation strength determined the rate with which naïve T cells initiated transcriptional activation but that cells activated by all ovalbumin-derived ligands were proliferating and expressing the effector molecule CD44 by two days (*Richard et al., 2018*).

Stimulated cells were stained with metal-conjugated antibodies and markers for dead cells and DNA before profiling by mass cytometry. We gated events detected by the mass cytometer in a hierarchical manner to select single, living cells that were TCRβ[+] and CD8α[+] before examining individual signalling molecules (*Figure 2a*). While gating for single cells based on DNA content (*Figure 2—figure supplement 2a*), we noted that a substantial percentage of events contained more than one cell-equivalent of DNA, particularly among cells stimulated with the strongest peptide, N4 (*Figure 2—figure supplement 2b*). Separating events by DNA content revealed two distinct populations, such that events containing two cell-equivalents of DNA had higher phosphorylated and total protein staining (*Figure 2b*, *Figure 2—figure supplement 2c–d*). It is therefore likely that this population contained multiple cells (i.e. doublets), potentially even cells engaged in TCR-peptide-MHC interactions with their neighbours. After normalization of each phosphorylated protein to the DNA signal detected in the same mass cytometry event, signal intensities for events with two cell-equivalents of DNA had similar ranges to those of events with one for most markers (*Figure 2—figure supplement 2e*). This provided further evidence that events with two cell-equivalents of DNA were doublets. Supporting the hypothesis that these doublets represented actively conjugated cells, a greater proportion of doublets than singlets showed signalling behaviour. In addition, from 1 to 4 hr after stimulation, pSLP76 signal was higher in doublet events even after normalization, suggesting that SLP76 was preferentially phosphorylated in cells actively engaged in TCR-peptide-MHC interactions. Because it was not possible to discern which proteins were signalling in which cell in a multiplet event, for subsequent analyses we included only singlet events.

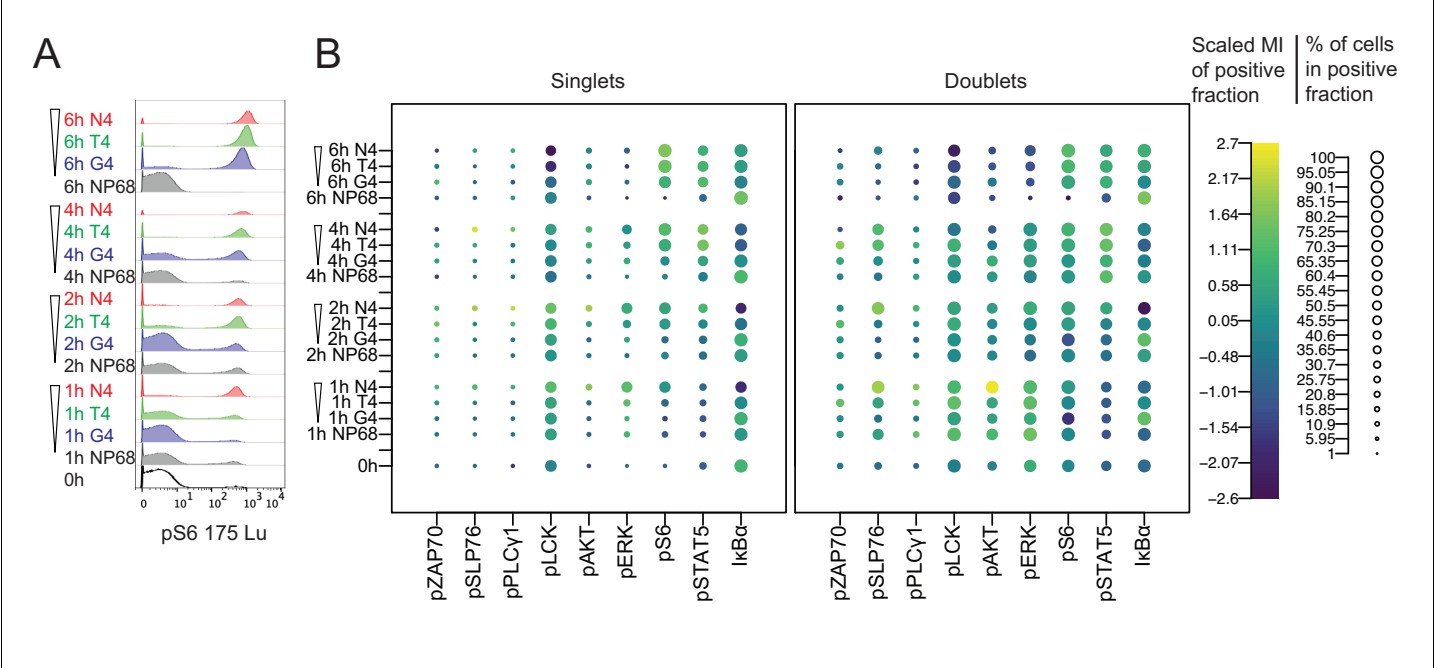

**Figure 2.** Mass cytometry measurements of signalling in singlet and doublet events. (a) Naïve CD8+ T cells were stimulated with 1 μM peptides of various potencies for 0, 1, 2, 4 and 6 hr before profiling by mass cytometry. Histograms depict pS6 signal. (b) Bubble plots of all signalling molecules after stimulation as in (a) in mass cytometry events with 1 or 2 cell-equivalents of DNA (singlets or doublets, respectively). The size of the bubbles denotes the percentage of positive cells, and the colour denotes the centred and scaled median intensity of each positive fraction. Results are representative of cells from six biological replicates measured in two independent experiments as detailed in *Supplementary file 1*.

The online version of this article includes the following figure supplement(s) for figure 2:

**Figure supplement 1.** Titrating peptide concentration.
**Figure supplement 2.** Gating based on DNA content and comparison of signalling markers.

## Ligand potency affects the kinetics of signalling protein activation

We next examined the kinetics of individual signalling molecules within our mass cytometry data. Total levels of signalling proteins did not substantially change over a 6 hr stimulation, while expression of effector proteins CD44 and CD25 increased in a time- and potency-dependent manner (*Figure 3—figure supplement 1*). In the presence of exogenous IL2, ligand potency did not strongly influence the rate with which individual cells phosphorylated STAT5 (*Figure 3—figure supplement 2a*). In the absence of exogenous IL2, STAT5 phosphorylation was associated with ligand potency, such that weak G4-stimulated cells showed no STAT5 phosphorylation (*Figure 3—figure supplement 2b*), likely due to autocrine/paracrine IL2 rapidly secreted by strongly stimulated cells (*Tan et al., 2017*). The percentages of cells degrading IκBα or phosphorylating S6 or ERK1/2 were not impacted by the presence of exogenous IL2. The percentages of cells phosphorylating pAKT [S473] were subtly increased by IL2 particularly under stimulation with low potency ligands (*Figure 3—figure supplement 2b*). This may reflect the mechanism proposed by Ross et al. whereby JAK signalling induced by IL2 ultimately stimulates mTORC2 phosphorylation of AKT[S473] (*Ross et al., 2016*). Together, these data indicate the selectivity of IL2 effects on T cell signalling pathways.

Phosphorylation of proximal, membrane-recruited mediators ZAP70 and PLCγ1 was only detectable in a small percentage of cells at any point during this time course, preventing further interpretation (*Figure 2b*). For LCK, the percentage of cells with inhibitory phosphorylation of pY505 decreased at 6 hr in a potency-dependent manner, suggesting that stronger stimuli resulted in greater LCK activity only at this late time point (*Figure 3—figure supplement 2a*). For SLP76, the high potency ligand N4 induced a greater percentage of signalling cells and greater signalling intensity within these cells between 1 and 4 hr, whereas signalling was minimal in cells stimulated with intermediate (T4) or low potency (G4) ligands (*Figure 2b*, *Figure 3—figure supplement 2a*).

Examination of the kinetics of individual distal signalling molecules revealed two distinct patterns. We defined these as transient if the percentage of signalling cells increased and subsequently decreased during the time course of high potency stimulation, and sustained if maximal signalling was ongoing at 6 hr (*Figure 3a*). ERK1/2, AKT and IκBα displayed transient signalling. While ERK1/2 and AKT are phosphorylated in response to TCR stimulation, IκBα is degraded, releasing NFκB subunits and permitting their translocation to the nucleus (*Paul and Schaefer, 2013*). Therefore, a reduction in IκBα$^+$ cells indicates active signalling by this node. For these three signalling mediators, the percentage of cells actively signalling was maximal at 2 hr when stimulated with the highest potency N4 peptide but was delayed until 4 or 6 hr when stimulated with the lower potency T4 or G4 peptides (*Figure 3a*). After 2 hr, signalling via these proteins declined in strongly stimulated cells, resulting in a convergence with more weakly stimulated cells. In addition, the maximum percentage of cells signalling through these nodes was substantially higher in strongly stimulated cells. This may indicate repeated node activation after high potency stimulation, such that a greater proportion of cells were signalling at any given time of measurement. Thus, for these transiently signalling proteins, ligand potency was associated with the maximal proportion of signalling cells and the speed with which this proportion was reached on a populational level.

In contrast to these transient signalling events, S6 phosphorylation induced by TCR stimulation was sustained within our time course (*Figure 3a*). Under N4 stimulation, there was a rapid initial increase in the percentage of pS6$^+$ cells before a plateau. This pattern may be indicative of the signalling protein approaching saturation. The appearance of pS6$^+$ cells was slower after stimulation with lower potency ligands, but the proportions of pS6$^+$ cells approached convergence between N4, T4 and G4 stimulations at 6 hr. Thus, for S6, the rate with which cells exhibited active signalling, but not the maximal proportion of signalling cells was associated with ligand potency.

We observed a bimodal distribution of pERK1/2 measurements in our mass cytometry data, consistent with previous reports (*Altan-Bonnet and Germain, 2005*; *Das et al., 2009*; *Tian et al., 2007*). The extent of ERK1/2 phosphorylation in pERK1/2$^+$ cells, as determined by the median marker intensity, was unaffected by ligand potency (*Figure 3—figure supplement 3a*). This confirmed that ERK1/2 exhibits digital signalling behaviour, that is on a per cell basis there is either an 'on' or 'off' state. Similarly, pS6[S235/S236] signal was also bimodally distributed (*Figure 2a*, *Figure 3—figure supplement 3b*). The intensity of pS6 signal in pS6$^+$ cells slightly increased over time. Normalization to total S6 protein intensity mitigated this effect (particularly under strong stimulation) and suggested that ligand potency may subtly affect the rate of S6 phosphorylation within individual cells (*Figure 3—figure supplement 3c*) in addition to the percentage of pS6$^+$ cells at early time points.

Since S6 phosphorylation at S235/S236 is driven by both S6K1 downstream of mTORC1, and RSK downstream of MEK1/2-ERK1/2 (*Pende et al., 2004*; *Roux et al., 2007*; *Salmond et al., 2009*), we were interested in how each of these pathways contributed to its digital behaviour in strongly stimulated cells. To this end, we treated cells with inhibitors of MEK1/2 (MEK162, *Lee et al., 2010*) and mTOR (rapamycin, *Pollizzi and Powell, 2015*) before stimulation with N4 peptides (*Figure 3—figure supplement 3d*). The per-cell phosphorylation of S6 decreased moderately in response to rapamycin (*Figure 3b–c*), with little difference between doses of 20 nM and 2 μM (*Figure 3—figure supplement 3e*), but the bimodal distribution of pS6 and the percentage of pS6$^+$ cells were not disturbed. In contrast, while S6 phosphorylation also decreased within each cell in response to MEK162, this response was dose-dependent between 0.5 and 5 μM, and at the highest dose (5 μM), the bimodality was disturbed. In addition, even low doses of MEK162 halved the percentage of cells with phosphorylated S6 (*Figure 3d*). Combined MEK162 and rapamycin resulted in severe inhibition of S6 phosphorylation (*Figure 3b*). Neither MEK162, rapamycin, nor the combination of these two substantially impacted CD44 expression at the concentrations of inhibitors used (*Figure 3—figure supplement 3f*), but a synergistic inhibition of cellular proliferation was observed after 2 days (*Figure 3—figure supplement 3g*). Thus, simultaneous activity of the MEK and mTOR pathways is required for phosphorylating S6[S235/S236] and proliferative responses, and MEK signalling is essential for S6 digital phosphorylation. These data emphasize the coordinated nature of signalling downstream of the TCR.

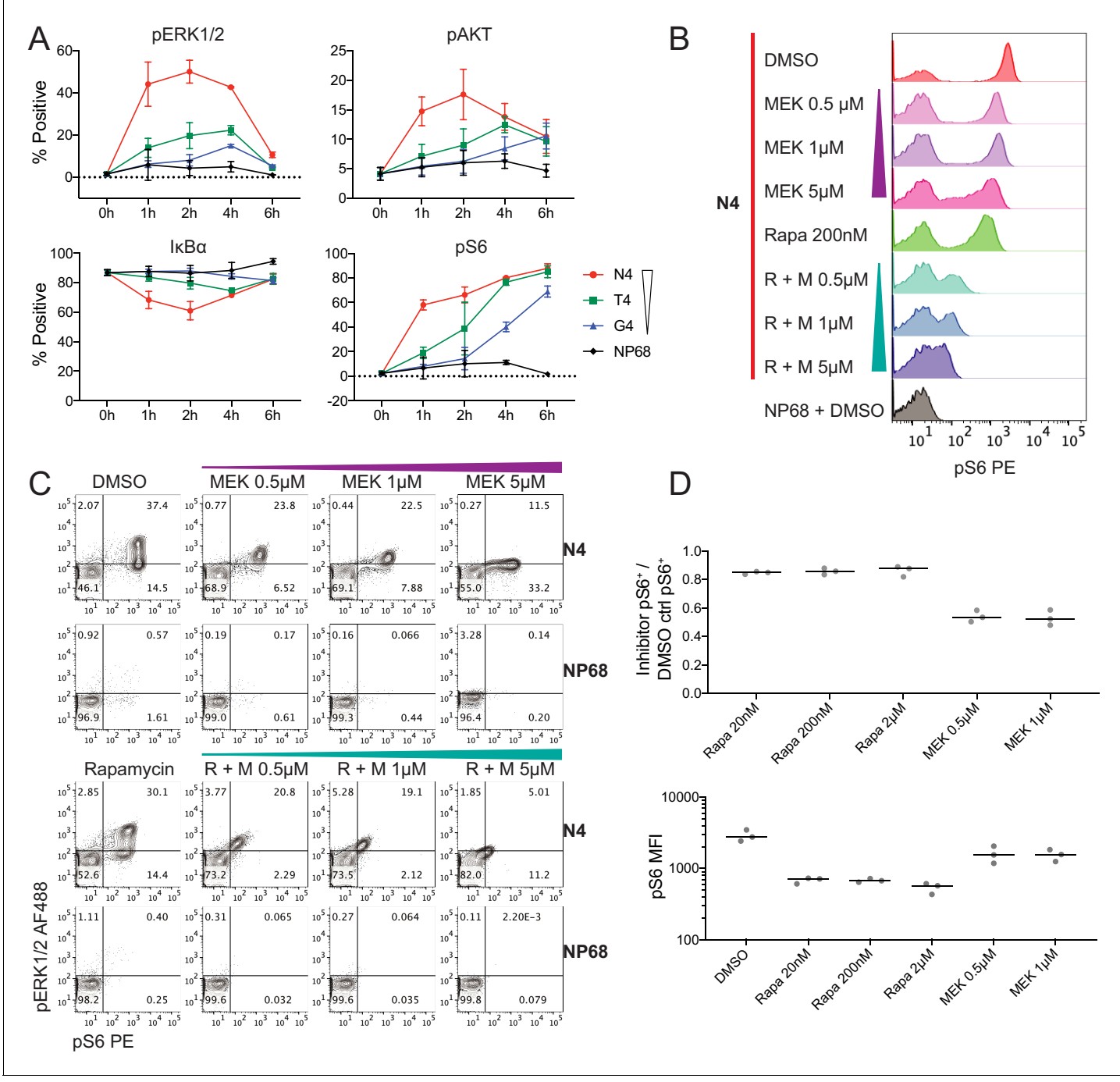

**Figure 3.** Kinetics of selected signalling proteins and impact of MEK and mTOR pathway inhibitors on T cell activation parameters. (a) The percentage of cells positive for each marker is plotted against time. Results depict combined data from six biological replicates measured in two independent experiments as detailed in *Supplementary file 1*. Points represent the mean and error bars depict the SD. Data underlying plots are provided in *Supplementary file 5*. (b–c) Flow cytometry measurements of pS6[S235/236] and pERK1/2 after 2 hr of pre-treatment with DMSO vehicle control, rapamycin (Rapa) 200 nM, MEK162 (MEK) 0.5 μM, 1 μM and 5 μM, or combined rapamycin with MEK162 (R + M), followed by 4 hr stimulation with 1 μM N4 or NP68 peptides. Results are representative of 3 independent experiments. (d) The fraction of pS6+ cells in N4-stimulated conditions with the indicated inhibitor treatments versus DMSO (top). The median fluorescent intensity of pS6 among pS6+ cells in N4-stimulated conditions with the indicated inhibitor treatments (bottom). Lines represent the median. Results depict combined data from three independent experiments.

The online version of this article includes the following figure supplement(s) for figure 3:

**Figure supplement 1.** Kinetics of total protein levels and surface markers.

**Figure supplement 2.** Kinetics of pSTAT5, pLCK and pSLP76 signalling proteins and testing the impact of IL2.

*Figure 3 continued on next page*

*Figure 3 continued*

**Figure supplement 3.** pERK and pS6 distributions and additional MEK and mTOR inhibition data.

## Ligand potency determines the abundance of signalling cells but not the coordination of signalling events

To take advantage of the simultaneous measurements in mass cytometry data, we next tested for differential abundance of multidimensional cellular phenotypes, taking into account all of the signalling markers measured (*Lun et al., 2017*; *Figure 4a*, *Figure 4—figure supplement 1*). We first defined 1585 fine-grained phenotypic states in the high dimensional space. We then compared the abundance of cells within these phenotypic states between unstimulated and all stimulated conditions (Materials and methods). Clustering of significantly differentially abundant states revealed two main signalling phenotypes. Phenotype A, defined as pS6$^+$ pSTAT5$^+$ pERK1/2$^+$, was most prevalent in the high potency (N4)-stimulated cells. Phenotype B, defined as pS6$^+$ pSTAT5$^+$ pERK1/2$^-$, appeared under all stimulation conditions with similar prevalence (*Figure 4a*). Phenotypes A and B were paralleled by the pSTAT5$^-$ phenotypes A' and B', respectively. Phenotypes A and A' were transient, whereas phenotypes B and B' were sustained. Subpopulation analysis confirmed that the high potency ligand N4 was capable of inducing a greater abundance of phenotypes A' (pS6$^+$ pSTAT5$^-$ pERK1/2$^+$) and A (pS6$^+$ pSTAT5$^+$ pERK1/2$^+$) up to 4 hr after stimulation (*Figure 4b*). In contrast, abundances of phenotypes B (pS6$^+$ pSTAT5$^+$ pERK1/2$^-$) and B' (pS6$^+$ pSTAT5$^-$ pERK1/2$^-$) increased at a similar rate between 1 and 6 hr after stimulation with all ovalbumin-derived ligands and were not associated with ordered ligand potency.

To complement these signalling phenotypes, we further investigated their relationship with surface expression of the effector protein CD44, which was an important contributor to phenotypic cluster separation (*Figure 4a*). We examined the 16 ($2^4$) possible states defined by combinations of the markers CD44, pS6, pSTAT5 and pERK1/2. The strongest peptide, N4, was capable of inducing a large proportion of CD44$^-$ pS6$^+$ pSTAT5$^-$ pERK$^+$ cells after 1 hr of stimulation (*Figure 4c*, *Figure 4—figure supplement 2*). This was accompanied by an increasing population of CD44$^-$ pS6$^+$ pSTAT5$^+$ pERK$^+$ cells, which reached its maximum abundance 2 hr after stimulation. Cells stimulated by the weaker peptides T4 and G4 showed dramatically reduced abundances of these cellular phenotypes, and their maxima were delayed. For all stimuli, a high abundance of CD44$^-$ pS6$^+$ pSTAT5$^{+/-}$ pERK1/2$^-$ cells was seen between 4 and 6 hr. Cells expressed CD44 by 4 hr after strong and intermediate stimulation (N4 and T4) and by 6 hr after weak stimulation (G4).

From these data, we inferred the coordination of distal TCR-induced signalling. We propose that within our stimulation system, activating cells initially phosphorylate S6 and ERK1/2, followed by STAT5, after which ERK1/2 becomes dephosphorylated, followed by STAT5 in some cells. In the early hours after stimulation, signalling cells express CD44 at the same level as unstimulated cells but begin upregulating CD44 expression by 4–6 hr. As time progresses and cells shift to sustained phenotypes, those activated with reduced potency ligands begin to phenotypically resemble those stimulated with high potency ligand (*Figure 5*). To formally test this order of activation events, we constructed activation trajectories of cells under each stimulation condition across all time points, based on their expression of pS6 (*Figure 5—figure supplement 1a*). We then asked in what order along these trajectories pS6, pERK1/2, pSTAT5, and CD44 activation events were initiated (*Figure 5—figure supplement 1b–c*). We found that pS6 appeared first, followed by pERK1/2, pSTAT5, and finally CD44. Of note, the start of ERK1/2 phosphorylation corresponded to the most dramatic increase in S6 phosphorylation, supporting evidence that ERK activation drives full S6[S235/S236] phosphorylation (*Figure 3*). The order of activation events was shared across stimulation conditions (p=0.00174 compared to random orders of events, Materials and methods). The signalling molecules pAKT, pLCK and IκBα were less dynamic along the trajectory, precluding precise determination of their order of activation particularly in weakly stimulated cells, but visualizing their changes along the trajectory further suggested shared patterns between stimuli (*Figure 5—figure supplement 1d*).

We therefore conclude that ligand potency controls the rate with which cells achieve certain signalling states and that the order of these signalling events is preserved regardless of stimulation strength.

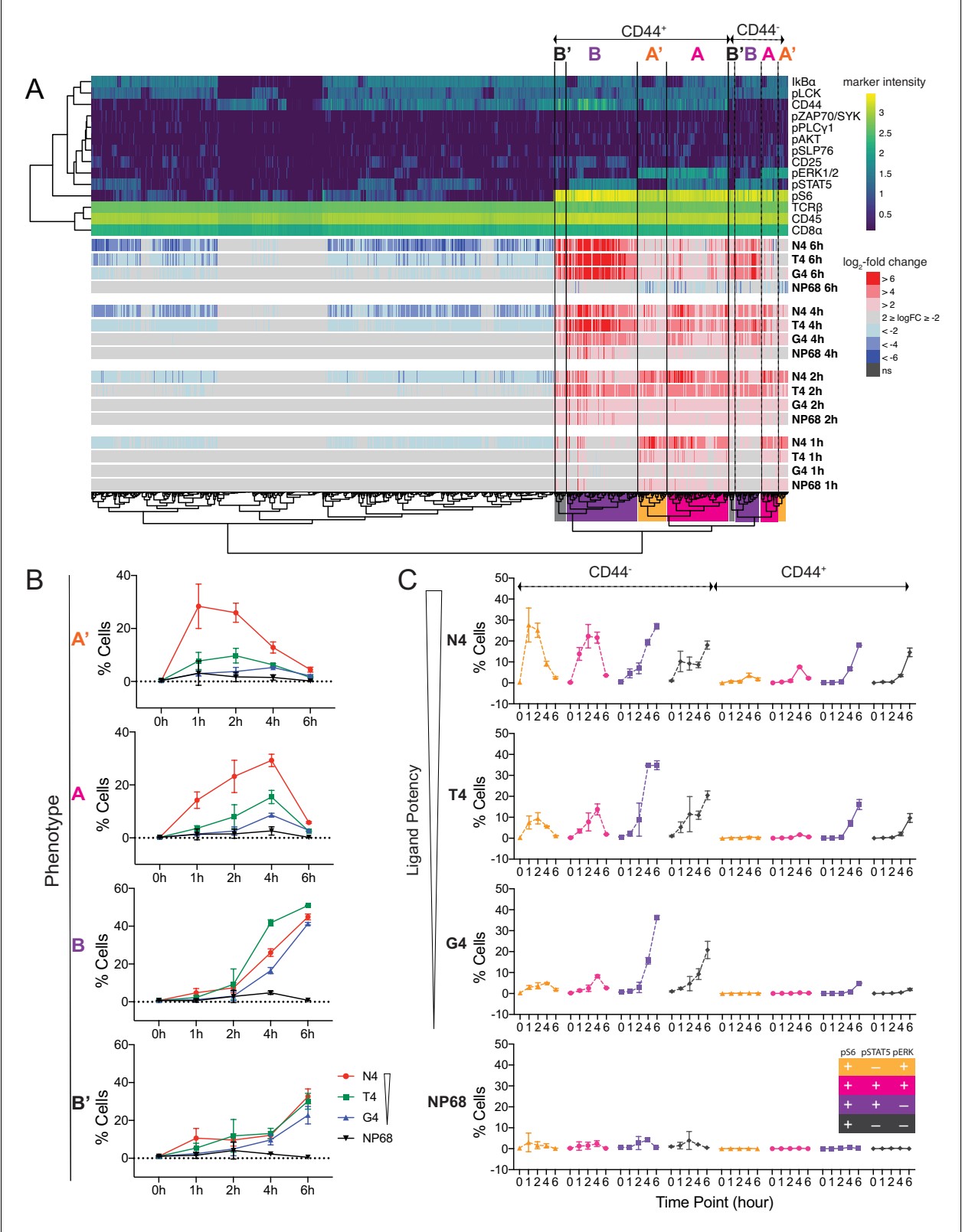

**Figure 4.** Examination of multi-dimensional phenotypes in mass cytometry signalling data. (a) Mass cytometry stimulation time courses were further investigated for multidimensional phenotypes that changed in abundance with stimulation. Analysis was run on two multiplexed biological replicates as described in Materials and methods. Phenotypic hyperspheres were defined within the multidimensional mass cytometry space and abundances of cells from each condition enumerated within each hypersphere. Each column in the heatmap represents an individual hypersphere. At the top of the

*Figure 4 continued on next page*

*Figure 4 continued*

heatmap, rows correspond to mass cytometry marker measurements with colour depicting the intensity of each marker in each hypersphere. Clustering by Pearson correlation was performed on these hypersphere marker intensity measurements. At the bottom of the heatmap, rows correspond to stimulation conditions with colour depicting the binned log$_2$-fold change in cellular abundance in stimulated versus unstimulated conditions within each hypersphere. ns = hyperspheres that did not significantly change in abundance. Phenotypic clusters of interest, A, A', B, B', are indicated in both CD44$^+$ and CD44$^-$ populations by coloured highlighting of the dendrogram. Statistics underlying the heatmap are provided in *Supplementary file 5*. (b) Percentages of cells exhibiting A, A', B, and B' phenotypes in mass cytometry measurements. Results are combined data from six biological replicates measured in two independent experiments as detailed in *Supplementary file 1*. Points represent the mean and error bars depict the SD. Data underlying plots are provided in *Supplementary file 5*. (c) Phenotypes as in (b) further sub-divided by CD44 expression.

The online version of this article includes the following figure supplement(s) for figure 4:

**Figure supplement 1.** t-SNE visualization of significantly differentially abundant hyperspheres.
**Figure supplement 2.** Heatmap of populations in *Figure 4c* including pS6$^-$.

## Biosynthetic pathways are coordinately regulated downstream of TCR activation

Finally, we shifted our focus further downstream to examine the relationship between signalling at the ribosomal protein S6 and mRNA expression of early response transcription factors. These two activation events indicate initiation of translational and transcriptional processes, which are required for the biosynthetic programs of T cell activation (*Araki et al., 2017*; *Howden et al., 2019*; *Tan et al., 2017*). S6 is a ribosomal protein whose phosphorylation reflects, though it does not regulate, TCR-induced translation (*Salmond et al., 2015*; *Salmond et al., 2009*). *Nr4a1* (Nur77) and *Irf8* encode transcription factors that are rapidly expressed upon T cell activation (*Moran et al., 2011*; *Nelson et al., 1996*), and we previously found that their transcripts are upregulated at 1 and 3 hr, respectively, after strong N4 stimulation (*Richard et al., 2018*; *Figure 6—figure supplement 1a*). To examine these translational and transcriptional characteristics simultaneously, we activated naïve OT-I CD8$^+$ T cells with ligands of various potencies before measurement of pS6 and mRNA molecules using combined phosphoflow and RNA flow cytometry (*Figure 6a*, *Figure 6—figure supplement 1b*).

Stimulation time courses with the different potency ligands suggested that *Nr4a1* transcripts were upregulated before phosphorylation of S6 and downregulated after, while *Irf8* transcripts were

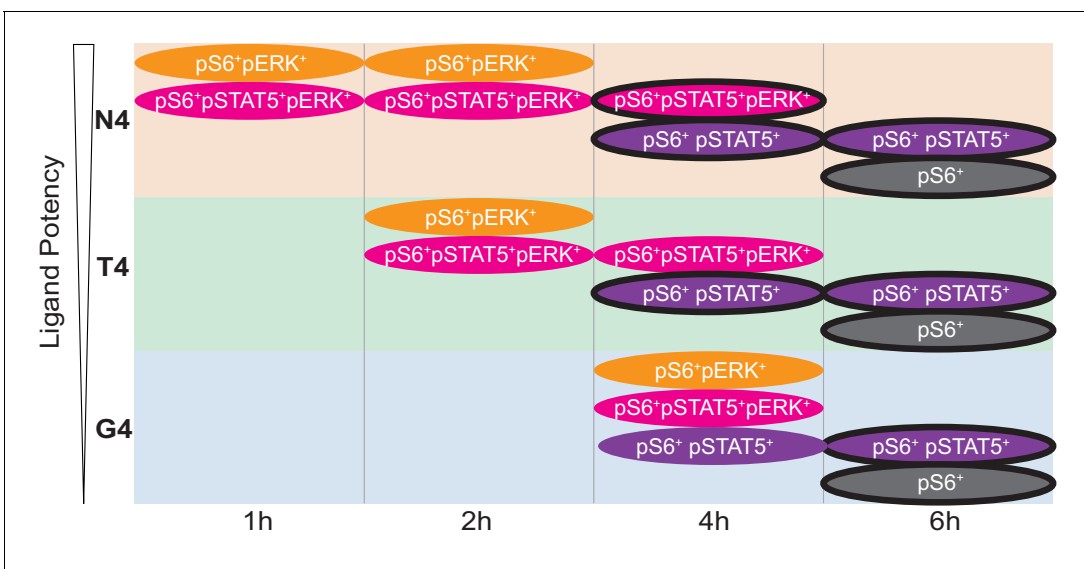

**Figure 5.** Cartoon Model. Cartoon depicts the kinetics of the four main signalling phenotypes in cells stimulated with ligands of varying potencies (N4, T4, G4) over time from data in *Figures 3a* and *4* (note the transient pERK$^+$ populations, even with G4). Black outlines indicate CD44$^+$ populations.

The online version of this article includes the following figure supplement(s) for figure 5:

**Figure supplement 1.** Activation trajectories for examining the order of signalling events.

upregulated after S6 phosphorylation (*Figure 6b*, *Figure 6—figure supplement 1c*). This order of events appeared consistent across stimuli. The percentage of pS6+*Nr4a1*+ cells was maximal between 1 and 2 hr after stimulation with the highest potency peptide N4, after 4 hr stimulation with the intermediate potency peptide T4, and after 6 hr stimulation with the lowest potency

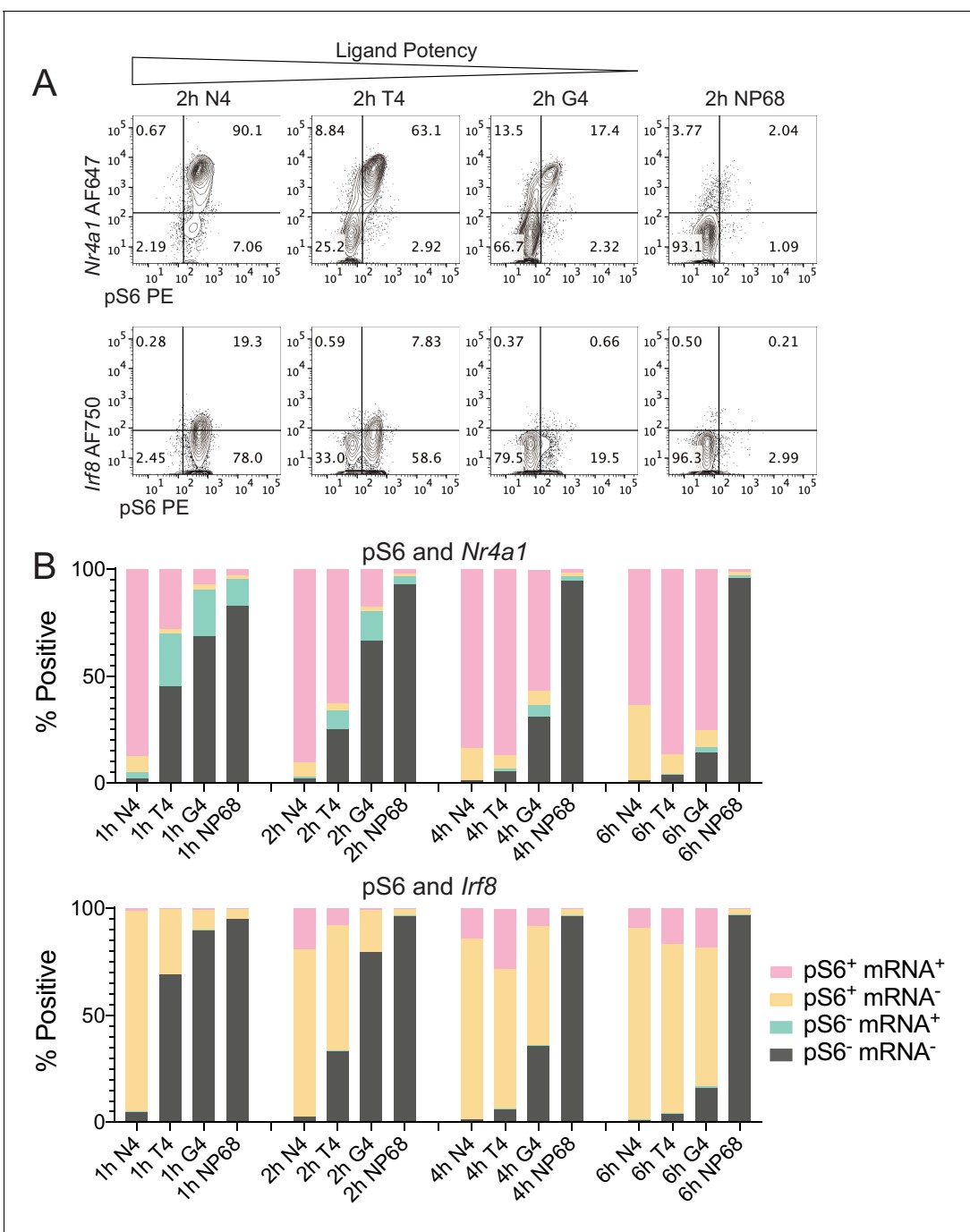

**Figure 6.** Simultaneous measurement of phosphorylation of S6 and mRNA expression of transcription factors Nr4a1 and Irf8. (a) Combined phosphoflow cytometry of pS6 and RNA flow cytometry of *Nr4a1* and *Irf8* transcripts in naïve OT-I CD8+ T cells stimulated with N4, T4, G4 or NP68 peptides for 2 hr, gated on single live cells in which the control gene *Rpl39* was detected. (b) Frequency of phenotypes depicted in (a) after stimulation for 1, 2, 4 or 6 hr. Data are representative of 3 independent experiments.

The online version of this article includes the following figure supplement(s) for figure 6:

**Figure supplement 1.** RNA flow cytometry gating strategy and histograms.

peptide G4. Likewise, the percentage of pS6$^+$Irf8$^+$ cells was maximal after 2 hr stimulation with N4, 4 hr stimulation with T4 and 6 hr stimulation with G4 peptides (*Figure 6b*). Similar to the multidimensional signalling phenotypes we measured by mass cytometry, these altered kinetics of phosphorylation and transcript upregulation indicate that stimulation strength controls their rate of activation.

These results suggest that the relationship between signalling events is conserved under different strengths of stimulation, even among the differing signal transduction pathways controlling transcription and translation. Upon TCR activation both the transcriptional and translational machinery are deployed in a coordinated manner, which may improve efficiency of protein production enabling the naïve CD8$^+$ T cell to differentiate and proliferate.

## Ligand potency affects the kinetics of selected signalling proteins in the presence of professional antigen-presenting cells

The interaction of adhesion molecules LFA-1 and ICAM-1 assists the formation of a stable immunological synapse, augments TCR-induced signalling, and continues to promote differentiation even after initial activation (*Gérard et al., 2013*; *Verma and Kelleher, 2017*). LFA-1 is constitutively expressed by naïve T cells, and TCR stimulation drives both redistribution and conformational changes that enhance its binding to the ligand ICAM-1 (*Capece et al., 2017*; *Dustin and Springer, 1989*; *Verma and Kelleher, 2017*). Palmer et al. previously demonstrated that LFA-1-ICAM-1 interactions improve conjugate formation during T cell stimulation with peptide-loaded splenocytes, particularly for low potency ligands (*Palmer et al., 2016*). However, it remained unclear whether ICAM-1 was expressed in our stimulation system and whether this integrin interaction could thereby play a role. We therefore measured ICAM-1 on the surface of T cells 6 hr after addition of pure peptides of various potencies (*Figure 7—figure supplement 1a*) and found that all T cells expressed ICAM-1, regardless of their stimulation status. These data suggest that integrin adhesion likely contributes to T cell activation along with TCR stimulation and exogenous IL2 in our system.

In contrast to this reductionist system, many additional factors impact T cell activation in vivo. Most fundamentally, naïve T cells are activated in the lymph node by professional antigen-presenting cells (APCs), such as dendritic cells, instead of other T cells. These APCs express costimulatory ligands in addition to peptide-MHC complexes, which can further tune naïve T cell responses (*Chen and Flies, 2013*; *Hubo et al., 2013*). For example, in our stimulation system, the costimulatory ligand CD80 remained largely absent after 6 hr of stimulation (*Figure 7—figure supplement 1a*). In contrast, mature bone marrow-derived dendritic cells (BMDCs) expressed high levels of CD80, along with additional costimulatory molecules (*Figure 7—figure supplement 1b–c*). To test how signalling responses to ligands of different strengths might be impacted by the additional signalling conferred by professional APCs, we stimulated naïve T cells with mature BMDCs loaded with peptides of various potencies. Exogenous IL2 was included to maintain comparability with the T:T stimulation system. Signalling molecules pZAP70, pSLP76, pERK1/2, pS6 and pSTAT5, as well as CD44 expression, were measured by flow cytometry.

We found that activation was in general less strong in the presence of peptide-pulsed professional APCs than pure peptides (*Figure 7*), perhaps due to reduced ligand availability as only half of the cells in the culture carried ligand (Materials and methods). The potency-dependent kinetics of pERK1/2, pSLP76 and CD44 resembled those observed in the T:T stimulation system, while pZAP70 remained undetectable. pS6 was upregulated over time under stimulation with high, medium and low potency ligands. pSTAT5 was upregulated over time with all stimuli, including the null peptide NP68, suggesting that simply mixing naïve T cells with BMDCs enhanced IL2 signalling. These results indicate that the rate-based mechanism we observed in the T:T stimulation system is further tuned at particular signalling nodes by more complex antigen presentation.

## Discussion

In this study, we examined the coordination of signalling pathways downstream of TCR activation using a custom mass cytometry panel as well as protein and RNA flow cytometry. The use of multidimensional measurements allowed us to probe the simultaneous activation of multiple signalling and transcriptional processes. This enabled comparisons of the impact of ligand potency on not only individual activation events but also their coordination. We found that the strength of TCR

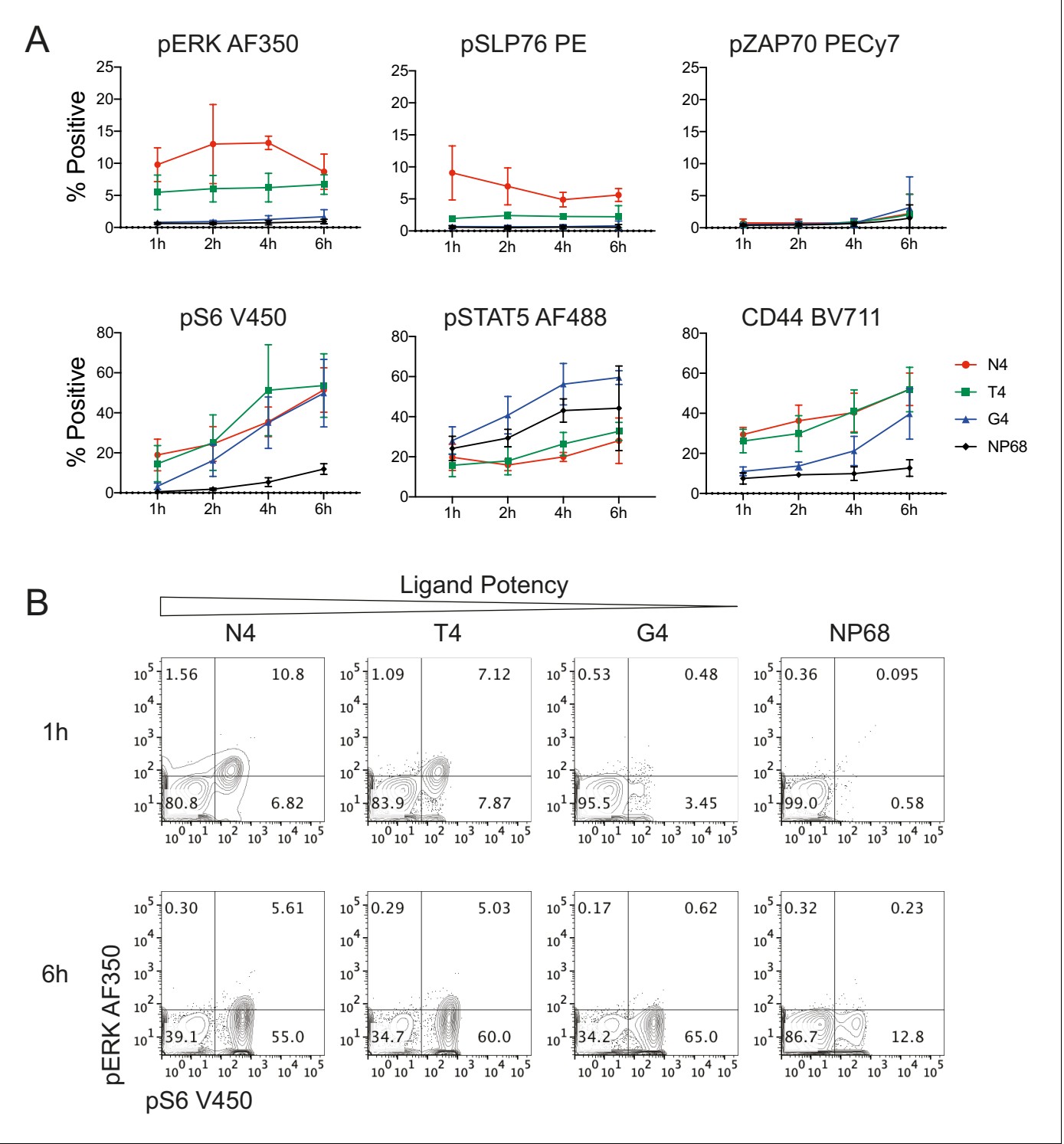

**Figure 7.** Signalling phenotypes in T cells stimulated with peptide-pulsed APCs. (a) Naïve CD8+ T cells were stimulated with mature BMDCs loaded with peptides of various potencies for 1, 2, 4 and 6 hr before profiling by flow cytometry. The percentage of cells positive for each marker is plotted against time. Results depict combined data from three independent experiments. Points represent the mean and error bars depict the SD. (b) Example flow cytometry data from (a) of pERK1/2 and pS6 measured at 1 and 6 hr.

The online version of this article includes the following source data and figure supplement(s) for figure 7:

**Source data 1.** Data underlying plots in *Figure 7a*.
**Figure supplement 1.** Expression of adhesion and costimulatory molecules on T cells and BMDCs.

stimulation controlled the rate of appearance of the multi-dimensional signalling and transcriptional phenotypes that we profiled.

Stimulation strength altering the rate of T cell activation has been observed in previous studies from our group and others investigating transcriptomic, proliferation, and protein characteristics (*Hommel and Hodgkin, 2007*; *Richard et al., 2018*; *Rosette et al., 2001*). Taken together, these data suggest that by controlling the probability that a cell will initiate activation responses, signal strength can modulate the average speed and magnitude of a population response. Our signalling results indicate that if such an activation switch exists, it lies very proximal to the TCR.

An important outstanding question is the mechanism by which the TCR translates ligand strength into the probability of downstream signalling. One model explaining the threshold for T cell response that could propagate to a rate-based mechanism is kinetic proofreading (*McKeithan, 1995*). This theory postulates that the ligand must remain bound to the receptor for sufficient time for signalling accumulation to surpass a critical event and propagate downstream. Indeed, multiple reports have suggested that naïve T cells require sustained interaction with presented antigen to achieve optimal proliferation, though the necessary duration differs by study and likely depends on the presence of additional factors such as IL2 (*Balyan et al., 2017*; *Curtsinger et al., 2003*; *Iezzi et al., 1998*; *Kaech and Ahmed, 2001*; *Prlic et al., 2006*; *van Stipdonk et al., 2003*; *van Stipdonk et al., 2001*; *Wong and Pamer, 2001*). Refinements to the kinetic proofreading model suggest that not a single interaction but rather the cumulative interaction lifetime of a series of early binding events controls signal accumulation (*Dushek et al., 2009*; *Liu et al., 2014*). Biophysical investigations of the impact of force on binding events between the TCR and pMHC (as well as CD8) have described catch bonds formed with high potency ligands that extend interaction lifetimes (*Das et al., 2015*; *Hong et al., 2018*; *Liu et al., 2014*; *Sibener et al., 2018*; *Wu et al., 2019*), though this observation has not been universal (*Limozin et al., 2019*) and merits further investigation. Extrapolating from this lifetime theory, altered ligand potency could change the probability of long or rapidly repeated binding events, thereby controlling the probability that an individual cell activates.

If such a lifetime-based mechanism exists, T cells must then translate variation in binding lifetime to the presence or absence of downstream signalling. Palmer and colleagues found ligand potency-associated differences in CD3ζ chain and ZAP70 phosphorylation (*Daniels et al., 2006*; *Palmer et al., 2016*), which may allow potency-dependent accumulation of signal before propagation downstream. Supporting this hypothesis, John James demonstrated that the number of CD3 ITAM motifs in a synthetic receptor influenced the rate but not the magnitude of signalling within individual Jurkat cells (*James, 2018*). Likewise, Mukhopadhyay et al. found that the presence of multiple ζ chain ITAMs, as well as ZAP70, increases the efficiency of phosphorylation in a HEK 293T cell reconstitution system, although these phosphorylation events do not account for the apparent switch-like ultrasensitive behaviour of T cell signalling (*Mukhopadhyay et al., 2016*). One mechanism that could explain the switch-like behaviour is the zero-order ultrasensitivity model (*Ferrell and Ha, 2014*; *Goldbeter and Koshland, 1981*), wherein negative regulators act in combination with activators to enhance responsiveness when signalling molecules operate close to saturation. In this way, a relatively small change in the binding lifetime of a pMHC ligand could be amplified by altering a kinase/phosphatase ratio to switch between the presence and absence of downstream signalling. An alteration in the local relative abundances of phosphatase CD45 and kinase LCK described by the kinetic segregation model (*Davis and van der Merwe, 2006*; *Razvag et al., 2018*) represents an intriguing candidate for controlling a zero-order ultrasensitivity mechanism (*Hui and Vale, 2014*), although a subsequent study has refuted its requirement for T cell activation (*Al-Aghbar et al., 2018*), suggesting other mechanisms. Control may also be mediated by the phosphatase PTPN22, which can dephosphorylate CD3ζ chains, ZAP70 and LCK (*Wu et al., 2006*), as absence of PTPN22 results in increased proportions of activated cells, particularly under weak stimulation (*Salmond et al., 2014*). Alternatively, Lo et al. showed that the slow phosphorylation of a tyrosine residue in LAT is a possible candidate for this rate-limiting step, since substitution of a single residue that enhances this phosphorylation improves T cell response to low potency ligands (*Lo et al., 2019*).

Through our single-cell measurements, we confirmed (*Altan-Bonnet and Germain, 2005*; *Das et al., 2009*; *Tian et al., 2007*) that ERK1/2 is phosphorylated with 'on or off' states, characteristic of digital signalling. In addition, we found that S6[S235/S236] is also phosphorylated in a similar

manner. However, whilst the extent of ERK1/2 phosphorylation during the 'on' state was constant, the extent of S6 phosphorylation subtly increased both with time and strength of stimulus. The parallel subtle increase in total S6 expression over time implies induction of S6 protein production during T cell activation. These data suggest that dividing signalling proteins into digital or analogue can be complicated by changes in total protein levels that may attribute analogue properties to a digital signal.

Previous work has shown that in addition to mTORC1 signalling, the MEK/ERK pathway contributes to S6[S235/S236] phosphorylation (*Pende et al., 2004*; *Roux et al., 2007*), particularly in naïve T cells (*Krishnaswamy et al., 2014*). Therefore, combinatorial effects of mTORC1 and MEK signalling might be expected to influence both S6 phosphorylation and other downstream T cell activation phenotypes. We found that chemical inhibition of both of these pathways blocked S6 phosphorylation, implying that they contribute in a non-redundant manner. A dose titration with MEK162 indicated that MEK/ERK signalling is critical for phosphorylation of S6[S235/S236]. Even at low doses, MEK162 reduced the percentage of pS6$^+$ cells, suggesting that it may modulate the rate of response. Furthermore, in our trajectory analysis of multiple signalling markers, a steep increase in S6 phosphorylation coincided with the appearance of pERK$^+$ cells under all peptide stimuli (*Figure 5—figure supplement 1a–b*). These data raise the possibility that digital phosphorylation of ERK propagates through RSK to S6[S235/S236].

We additionally explored the effects of rapamycin and MEK162 on naïve T cell proliferation. Although rapamycin had little effect on S6 phosphorylation, it had a profound effect on T cell proliferation, which may be due to several different mechanisms. First, rapamycin can also impact mTORC2 signalling after prolonged treatment (*Sarbassov et al., 2006*), and naïve T cells may be particularly susceptible (*Delgoffe et al., 2011*). Second, mTORC1 affects many additional pathways other than ribosomal activity (*Pollizzi and Powell, 2015*; *Salmond, 2018*). Finally, even when signalling through S6K1, mTORC1 can influence proliferation through pS6-independent mechanisms (*Salmond et al., 2015*). We also found that MEK inhibition with MEK162 synergized with rapamycin to further dampen T cell proliferation, highlighting the interconnected nature of the signalling pathways downstream of the TCR.

Many signalling molecules exhibited transient behaviour at the population level (ERK1/2, IκBα, AKT), while pS6 accumulated over the course of our 6 hr experiments. Stimulation strength strongly influenced the proportion of cells exhibiting transient signalling behaviours between 1 and 4 hr after activation, but by 6 hr, cells activated with any of the ovalbumin-derived peptide ligands exhibited a similar signalling phenotype. This potency-dependent difference in the maximal proportions of cells signalling may be due to either repeat or sustained signalling with strong ligands, the latter of which has been observed for calcium fluxes induced by TCR stimulation (*Chen et al., 2010*; *Le Borgne et al., 2016*; *Wülfing et al., 1997*). For example, although under weak G4 stimulation only a very small percentage (15.4%) of cells were found to be pERK1/2$^+$ at any given time, the majority (72.1%) of G4-stimulated cells achieved digital activation of pS6[S235/236] by 6 hr (*Figure 3a*). Given that we found full S6 phosphorylation after strong stimulation requires MEK signalling, we hypothesize that this pathway is active in all stimulation conditions but that ERK activation events occur with reduced frequency or duration with weak stimuli and thus many were missed in our snapshot measurements. Future investigations using ERK reporters and ERK inhibition in weakly stimulated cells would be needed to test this prediction. Consistent with this proposed mechanism, single-cell studies in epithelial and HEK293 cell lines have observed oscillating ERK phosphorylation with frequency and duration dependent on the concentration or frequency of EGF stimulation (*Albeck et al., 2013*; *Ryu et al., 2018*). Such an effect on digital ERK activation may be modulated by multi-step activation of the upstream mediator SOS dependent on its dwell-time after activation-induced recruitment to the plasma membrane (*Huang et al., 2019*). Interestingly, using a light-inducible ERK activation system in epithelial cells, Aoki et al. demonstrated divergent transcriptional effects of sustained versus transient ERK activation (*Aoki et al., 2013*). It therefore remains possible that different ERK targets in T cells, such as translational machinery, microtubule remodelling, and transcription factors (e.g. ELK1, SAP1, SAP2) (*Navarro and Cantrell, 2014*) are differentially affected by stimulation strength, warranting further investigation of additional downstream components.

Examination of the coordinated activation of transcriptional and translational signalling pathways also revealed conservation of this order of events. Biosynthetic processes are critical for naïve T cells

to differentiate into effector cells (*Araki et al., 2017*; *Tan et al., 2017*), and thus, carefully controlled simultaneous activation would ensure efficient, consistent effector differentiation of activated cells.

Under stimulation with peptide-loaded BMDCs, ligand potency determined the percentages of T cells undergoing certain activation events (pERK1/2, pSLP76 and CD44), similar to observations in our reductionist stimulation system. In contrast, phosphorylation of S6 was not associated with ligand potency after stimulation with peptide-loaded BMDCs. Unlike naïve and recently activated T cells, BMDCs express high levels of costimulatory molecules that can impact TCR-induced signalling. For example, ligation of the costimulatory receptor CD28 at the same time as the TCR results in amplification of signalling pathways including NFAT, NFκB and AP-1, and can enhance both the sensitivity and ultimate division potential of naïve T cell activation (*Esensten et al., 2016*; *Heinzel et al., 2017*; *Marchingo et al., 2014*; *Yang et al., 2017*). Further exploration of how individual costimulatory ligands impact the coordination and initiation rate of the TCR-induced signalling programme will be important for dissecting these additional inputs.

Despite the increased complexity of BMDC peptide presentation, this in vitro system is nevertheless still far-removed from in vivo T cell activation, where the microenvironment is increasingly complex. Additional variables such as cytokine and nutrient availability and cell-cell interactions can further tune the T cell response in vivo (*Curtsinger and Mescher, 2010*; *Kedia-Mehta and Finlay, 2019*). Moreover, strongly stimulated T cells undergo prolonged retention in the lymph node (*Ozga et al., 2016*; *Zehn et al., 2009*) and may out-compete weakly stimulated T cells for cytokines and nutrients (*Wensveen et al., 2012*), suggesting that stimulation strength and the microenvironment are not independent. Our controlled in vitro systems allowed us to identify effects of stimulation strength on TCR-induced pathways alone, as well as in the context of BMDC-mediated costimulation, without confounding by other in vivo factors and feedback. By delineating the impact of stimulation strength in low-complexity systems, these data can form the basis for interpretation of future studies where additional variables may be explored.

In this study, we measured 22 markers of protein expression and active signalling. While other unmeasured signalling mediators may respond to altered stimulation strength in a different manner, our data demonstrate a strict choreography of the distal signalling processes that we examined. Stimulation strength was associated with the rate with which cells embarked on this regimented programme. This suggests that using a limited set of signalling machinery in a single coordinated programme, T cells can finely tune their responses to different ligands through modulation of the rate of signalling.

## Materials and methods

### Key resources

Key resources are detailed in *Supplementary file 2*.

### Mice

CD8[+] T cells were isolated from OT-I *Rag1*-deficient mice (OT-I *Rag1*[tm1Bal] on a C57BL/6 background), which underwent confirmation of genotype prior to study. BMDCs were generated from wild-type C57BL/6 mice. Experiments used both male and female mice 9–25 weeks old. Mice were bred and maintained within University of Cambridge animal facilities.

### Cell culture and stimulation

For T cell isolation, single cell suspensions of splenocytes were produced via homogenization of the spleen through a 70 μM nylon strainer. CD8α[+] T cells were isolated using the Mouse CD8α[+] T cell Isolation Kit (MACS Miltenyi Biotec). Cells were cultured in RPMI 1640 (Gibco), 10% FBS (Biosera), penicillin-streptomycin (Sigma), sodium pyruvate (Gibco), L-glutamine (Sigma), β-mercaptoethanol (Gibco) and 20 ng/ml recombinant mouse IL-2 (Peprotech). For stimulation, the following peptides were used at the concentrations indicated: SIINFEKL (N4), SIITFEKL (T4), SIIGFEKL (G4), and ASNENMDAM (NP68) (Cambridge Bioscience).

Bone marrow derived dendritic cells (BMDCs) were generated based on a published protocol by Abcam. Femurs and tibias were sterilized in 70% ethanol and flushed with cold BMDC culture media, consisting of RPMI 1640 (Gibco), 10% FBS (Biosera), penicillin-streptomycin (Sigma), L-glutamine

(Sigma) and β-mercaptoethanol (Gibco). The suspension of bone marrow progenitor cells was passed through a 70 µM nylon strainer and plated in 10 cm petri dishes in BMDC culture media supplemented with 20 ng/ml GM-CSF (Peprotech). Fresh BMDC culture media with 20 ng/ml GM-CSF was added on day three and replaced on day six and, if needed, day 8. Immature dendritic cells were harvested from day 7 to day 9. Maturation was induced by culturing immature dendritic cells for 1 day in BMDC culture media with 20 ng/ml GM-CSF, as well as 50 ng/ml LPS (Thermofisher Scientific) and 20 ng/ml IL4 (Abcam). Differentiation into immature and mature BMDCs was verified by flow cytometry. To stimulate T cells, mature BMDCs were pulsed with 1 µM of peptide for 1 hr at 37° C, washed, mixed with naïve T cells at a ratio of 1:1, and cultured for the times indicated.

## Mass cytometry

Purified naïve CD8+ T cells were analysed by mass cytometry. In experiment 1, cells from four age-matched mice (two males and two females) were used, representing four biological replicates. In experiment 2, more stimulation conditions were included. This necessitated more cells for each biological replicate than could be obtained from a single mouse. Therefore, each biological replicate (one male, one female) was composed of cells from a pair of age- and gender-matched mice. Staining for mass cytometry was performed using sequential MaxPar reagent kits (Fluidigm) in the following steps. Live cells were stained with 5 µM Cell-ID Cisplatin for 5 min at 37°C and rested for 15–30 min before stimulation with 1 µM N4, T4, G4, or NP68 peptides, or left unstimulated. In Experiment 1, cells were stimulated for 1 and 2 hr. In experiment 2, cells were stimulated for 1, 2, 4 and 6 hr. See *Supplementary file 1* for replicate structure. Cells were fixed with Maxpar Fix I Buffer for 10 min at room temperature. Cells stimulated under different conditions were barcoded using the Cell-ID 20-Plex Pd Barcoding Kit and pooled for staining to minimise confounding technical differences. In experiment 1, all cells from each mouse were pooled into a batch. In experiment 2, in addition to pooling within a biological replicate, four samples were shared across the two pools to enable batch normalization for differential abundance analysis as described below. Cells were blocked with FCR blocking reagent (Biolegend, clone 93) and stained with metal-conjugated surface antibodies (*Supplementary file 3*). Surface-stained cells were permeabilized with methanol (Fisher Scientific) and stained with metal-conjugated antibodies against intracellular targets (*Supplementary file 3*), all diluted in Maxpar Cell Staining Buffer. Stained cells were then fixed with 1.6% formaldehyde (Thermofisher) and stained overnight with 125 nM Cell-ID Intercalator-Ir in Maxpar Fix and Perm Buffer. Cells were analysed on a Helios CyTOF system (Fluidigm). Data within each cell pool were normalized and debarcoded using the Fluidigm CyTOF software.

## Mass cytometry antibodies

Metal-conjugated antibodies were custom-conjugated where not already commercially available (*Supplementary file 3*). All custom-conjugated antibody clones were tested using phosphoflow cytometry before and after metal-conjugation. When allocating metals to antibody targets, brighter metals were assigned to antibodies that exhibited weaker phosphoflow staining or to those without clear bimodal expression. Metal channels that receive significant cross-over from other channels were also allocated antibodies with stronger signals. For each protein target, antibodies against the total protein and its phosphorylated version were conjugated to metals differing by more than one mass unit to avoid spillover.

Antibodies targeting phosphorylated proteins were validated using phosphoflow (with and without metal conjugation) under different stimulating conditions, including anti-CD3 coated plate (1 µg/ml, BD Biosciences, clone 145–2 C11), PMA (50 nM, Sigma-Aldrich) and ionomycin (1 µg/ml, Sigma-Aldrich), N4 peptide (1 µM), and pervanadate (1 mM, prepared using sodium orthovanadate, Sigma-Aldrich) (*Supplementary file 4*). To confirm the specificity of the antibody clones targeting ZAP70, LCK and SLP76, we transfected HEK 293 T cells, which lack endogenous expression of these proteins, with vectors encoding the proteins and tested antibody binding via flow cytometry and immunofluorescence (*Supplementary file 4a*). To confirm the specificity of the antibody clone targeting PLCγ1 (3H1C10), we performed siRNA knockdown in T cells and tested antibody binding by flow cytometry (*Supplementary file 4a*). (Knockdown was validated by western blotting with a WB-specific antibody clone (D9H10).) Further validation of phospho-specific antibodies was performed using signalling inhibitors as detailed in the antibody specificity tables (*Supplementary file 4*).

All metal-conjugated antibodies were tested by mass cytometry prior to experimentation. By testing antibodies on fixed cells, fixed and barcoded cells, and live cells (surface markers only), we confirmed there was no additional loss of antibody activity through the addition of the barcoding step. Two surface markers performed less well when fixed (CD62L-160Gd clone MEL-14 and CD69-143Nd clone H1.2F3, both Fluidigm). The CD62L-160Gd antibody was excluded from the mass cytometry panel. The CD69-143Nd antibody was excluded from the analyses in experiment 1, and excluded from the staining panel in experiment 2. All metal-conjugated antibodies were titrated for optimal performance and key signalling antibodies were tested in a time-course assay under different stimulatory conditions to determine the optimal times for running the full panel.

## Mass cytometry data analysis

For mass cytometry analysis in FlowJo (v10), debarcoded samples were gated in a hierarchical manner: EQ bead exclusion followed by selection of intact cells based on DNA content, single cells based on the event length and DNA content, living cells based on cisplatin staining, and finally CD8$\alpha^+$ TCR$\beta^+$ cells. For activation-induced markers, positive/negative status was defined based on comparison with unstimulated cells.

Normalization of antibody-measured signals to DNA signal, as well as phospho-protein to total protein signals, was performed in R using the ncdfFlow (v2.30.1) (*Gopalakrishnan, 2019*) and flowCore (v1.50.0) (*Hahne et al., 2009*) Bioconductor packages. The signal of each marker in each event was normalized to the signal from the 191Ir DNA channel or appropriate total protein channel within that event. Normalized ratios were then scaled to the median 191Ir DNA or appropriate total protein signal from one selected sample for visualization purposes.

To test for differential abundances, mass cytometry data from experiment 2 was processed using the ncdfFlow (v2.30.1), flowCore (v.1.50.0) and cydar (v1.8.0) (*Lun et al., 2017*) Bioconductor Packages in R. A logicle transformation (default parameters except w = 0.1) was applied to raw intensity data. Data from the two batches were range-normalized based on the four samples that were included in both batches using the normalizeBatch function (with parameters p=0.001, fix. zero = TRUE). After normalization, one technical replicate from each of these four repeated samples was carried forward for analysis. All samples were then pooled before constructing the sequential gating strategy: removal of residual EQ beads, removal of events with high event length, retaining events with a single cell-equivalent of DNA, removal of dead cells, removal of cells with TCR$\beta$ signal more than 5 MAD below the median, and removal of cells with CD8 signal more than 5 MAD below the median. Cells from each sample were down-sampled to the number in the smallest sample (10,982). Only signalling proteins and selected surface markers were included in differential abundance testing (to avoid invariant and non-biological markers): pSTAT5, pAKT, pSLP76, pLCK, I$\kappa$B$\alpha$, pPLC$\gamma$1, pERK1/2, pZAP70, pS6, CD8$\alpha$, CD44, CD25, TCR$\beta$, CD45.

To test for differential abundance of cells with any combinatorial phenotype, agnostic to cellular density or clustering patterns, we employed cytometry differential abundance testing in R (cydar, *Lun et al., 2017*). This method takes advantage of the consistent staining achieved with sample barcoding, along with the count-based nature of single cell data, to find regions of the high-dimensional marker space occupied significantly more or less frequently by cells from a particular condition. This is achieved by filling the marker space with hyperspheres, comparing cellular abundances within each hypersphere across conditions, and controlling the false discovery rate across the marker space. Cells were assigned to hyperspheres and counted using the prepareCellData, neighborDistances, and countCells functions from cydar (default parameters except countCells tol = 0.4 and downsample = 200). Hyperspheres were included in differential abundance analysis if they contained more than 50 cells on average. Differential abundance was assessed using the edgeR (v3.26.8) Bioconductor package (*Lun et al., 2016*; *Robinson et al., 2010*) with a robust quasi-likelihood GLM fit (*Lun et al., 2017*) including the biological replicate of origin as a blocking factor for each sample in an analysis of deviance test to identify hyperspheres that changed in abundance in any stimulation condition compared to the unstimulated control. The spatial FDR was controlled at 0.05 to define significantly differentially abundant hyperspheres. See *Supplementary file 5* for full summary statistics from differential abundance testing.

For trajectory analysis, each biological replicate was analysed separately. A logicle transformation (default parameters except w = 0.1) was applied to raw intensity data. Cells within each replicate were then gated using the sequential strategy described above. The MAD threshold for TCR$\beta^+$ cells

was relaxed in gating biological replicates from experiment 1 due to a wider distribution in this dataset. To construct trajectories, equal numbers of cells stimulated by each ovalbumin-derived ligand were pooled with unstimulated cells. This created one sample per ligand (N4, T4 and G4) per biological replicate from which to construct trajectories. For each trajectory, cells were ordered by intensity of pS6 as this marker was observed to increase with activation over real time. Colouring cells by real time point confirmed enrichment of cells sampled at early times at the beginning of the trajectory and later times at the end (*Figure 5—figure supplement 1a*). To generate plots in *Figure 5—figure supplement 1b–c*, a loess curve was fitted to intensity measurements of the indicated markers across 2000 randomly sampled cells from each trajectory (span = 0.2). To determine the trajectory interval in which each activation event started, trajectories were downsampled to 5000 cells each, and a sliding window encompassing 5% of the trajectory was established to move across the trajectory from least to most activated in steps of 1%. The first window in which the mean intensity of cells was more than one standard deviation away from the mean intensity in the starting window was deemed the initiation of the activation event. Events that displayed a shift in mean intensity across the trajectory but fell short of the threshold (CD44 under G4 stimulation), were considered to be last in the ordering. If more than one activation event failed to meet this threshold in a given trajectory, or if two events shared an initiation window, it was not possible to robustly declare the order. We then computed the probability that orders of signalling events would be shared between each pair of trajectories to the observed extent or more if orders were random. To do this, we compared the mean-squared-distance (MSD) between the orders in trajectory 1 and trajectory 2 to a distribution of MSDs between the orders in trajectory 1 and permuted orderings of trajectory 2. Both biological replicates from experiment 2 that contained all time-points revealed identical orders of activation events across all stimulation conditions. The two biological replicates in experiment 1 that included 0, 1 and 2 hr of stimulation also revealed the same order of activation of pS6 and pERK, while pSTAT5 and CD44 were not sufficiently activated by 2 hr to determine their ordering. It was not possible to order events in the remaining biological replicates from experiment one that included only one stimulated timepoint.

## Flow cytometry

To test BMDC maturation, cells were stained with live-dead marker (Zombie-NIR or Zombie-Aqua Fixable Viability Kit, Biolegend) in PBS before staining in incubation buffer (1% FBS in PBS) with FCR (FC receptor) blocking antibody (Biolegend, clone 93) and antibodies against CD11c, MHC II, CD80, CD86, CD40 and ICAM1 (*Supplementary file 2*). Cells were acquired on a BD LSRFortessa. Data were analysed in FlowJo (v10) gating on single, live cells. Mature BMDCs were consistently >90% CD11c$^+$ and MHC II$^+$ (*Figure 7—figure supplement 1b*). To measure CD80 and ICAM-1 expression on activated T cells, T cells were stained with live-dead marker (Zombie-NIR Fixable Viability Kit, Biolegend) and antibodies against CD80 and ICAM1 in the same way.

To test the impact of inhibiting the MEK and mTOR pathways on cell proliferation (*Figure 3—figure supplement 3g*), cells were stained with eBioscience Cell Proliferation Dye eFluor-450 (Thermo-Fisher), pre-treated for 2 hr with MEK162 (1 μM and 5 μM), rapamycin (200 nM) or combined MEK162 (1 μM or 5 μM) and rapamycin (200 nM), and stimulated with 1 μM N4 or NP68 peptides for 2 days. Cells were then stained with live-dead marker (Zombie-NIR Fixable Viability Kit, Biolegend) and acquired on a BD LSRFortessa. Data were analysed in FlowJo (v10) gating on live, single cells.

## Phosphoflow cytometry

For phosphoflow cytometry experiments in *Figure 2—figure supplement 1*, *Figure 3b–d*, *Figure 3—figure supplements 2b* and *3d–g*, and *Figure 7*, after stimulation, cells were fixed in 4% paraformaldehyde (Electron Microscopy Sciences) at room temperature for 15 min and washed in PBS. Cells were permeabilized with 90% ice-cold methanol (Fisher Scientific) for 30 min on ice or overnight at −20℃. Cells were washed in PBS and resuspended in 100 μL incubation buffer containing FCR blocking antibody (Biolegend, clone 93), stained with the primary antibodies of interest (*Supplementary file 2*), and incubated for 1 hr at room temperature. In cases where the primary antibody was not conjugated to a fluorophore, the cells were then washed, resuspended in 100 μL incubation buffer containing FCR blocking antibody and secondary antibody (*Supplementary file 2*)

and incubated for 30 min at room temperature. Cells were washed in incubation buffer prior to data acquisition on a BD LSRFortessa. Data were analysed in FlowJo (v10) gating on single, live cells.

To test the impact of titrating peptides on the phosphorylation of ERK and S6 (*Figure 2—figure supplement 1*), cells stained with live-dead marker (Zombie-NIR Fixable Viability Kit, Biolegend) were stimulated with N4, T4, G4 and NP68 peptides at concentrations of 10 nM, 100 nM and 1 µM for 2 and 4 hr.

To test the impact of adding or withholding exogenous IL2 on phosphorylation of STAT5, S6, ERK, and AKT, and degradation of IκBα (*Figure 3—figure supplement 2b*), cells stained with a live-dead marker (Zombie-NIR Fixable Viability Kit, Biolegend) were stimulated with 1 µM of peptides for 4 hr.

To test the impact of inhibiting the MEK/ERK and mTOR/S6 pathways (*Figure 3b–d*, *Figure 3—figure supplement 3d-f*), cells stained with live-dead marker (Zombie-NIR Fixable Viability Kit, Biolegend) were pre-treated with the MEK inhibitor MEK162 (binimetinib/ARRY-162/ARRY-438162, Selleckchem), mTOR inhibitor rapamycin (Sigma-Aldrich) or combined MEK162 and rapamycin for 2 hr. MEK162 was added at 0.5 µM, 1 µM and 5 µM, rapamycin was added at 20 nM, 200 nM and 2 µM. For combined drug treatments, MEK162 was added at 0.5 µM, 1 µM and 5 µM with rapamycin at 200 nM. DMSO as a vehicle control was added at 1:1000, corresponding to the amount in the 200 nM dose of rapamycin and the 1 µM dose of MEK162. Cells were stimulated with 1 µM of N4 or NP68 peptides for 4 hr.

To test naïve T cell stimulation with peptide-loaded BMDCs, T cells were stained with live-dead marker (Zombie-NIR Fixable Viability Kit, Biolegend) before co-culture with BMDCs.

## Combined phosphoflow with RNA flow cytometry

To combine phosphoflow with RNA flow cytometry (*Figure 6* and *Figure 6—figure supplement 1b-c*), purified naïve CD8α⁺ T cells were stained using a live-dead marker (Zombie Aqua Fixable Viability Kit, Biolegend). To achieve a sufficient number of cells, isolated naïve CD8⁺ T cells from three age- and gender-matched mice were pooled for each biological replicate. Cells were stimulated for 0–6 hr with 1 µM N4, T4, G4, or NP68 peptides. At the end of stimulation, cells were immediately moved on to ice and washed with cold PBS. Cells were fixed and permeabilized using the Primeflow RNA Assay Kit (ThermoFisher Scientific), blocked with FCR blocking reagent (Biolegend, clone 93) and stained with antibodies against pS6[S235/236] (BD Biosciences clone N7-548) for 30 min. Cells were stained with the following PrimeFlow probe sets (Thermofisher Scientific): *Nr4a1* AF647 (Type1, VB1-12484-204), *Irf8* AF750 (Type 6, VB6-3197312-210), and *Rpl39* AF488 (Type 4, VB4-3120826-204) as a control. The use of *Rpl39* as a control gene was previously described in naïve and recently activated CD8⁺ T cells (*Richard et al., 2018*). Cells were acquired on a BD LSRForessa and analysed in FlowJo (v10). Cells were gated on single, live cells that expressed *Rpl39*, to ensure cells were permeabilized and probes hybridized and amplified.

## Code availability statement

Analysis code for mass cytometry data is available at https://github.com/MarioniLab/SignallingMassCytoStimStrength (*Ma, 2020*; copy archived at https://github.com/elifesciences-publications/SignallingMassCytoStimStrength/).

## Acknowledgements

This work was funded by the Wellcome Trust (grants [103930], [100140] and [217100] to GMG and grant 204017/Z/16/Z to CYM); Cancer Research UK (core funding [A17197] to JCM); EMBL (core funding to JCM); the MRC (MR/P014178/1 to ACR); the ACT (grant [23/17 A (ii)] to CYM). ACR also received a pump-priming grant from the National Institute for Health Research [Cambridge Biomedical Research Centre at the Cambridge University Hospitals NHS Foundation Trust]; the views expressed are those of the authors and not necessarily those of the NHS, the NIHR or the Department of Health and Social Care. This research was supported by the CIMR and CRUK-CI Flow Cytometry Core Facilities. We thank M Strzelecki and R Grenfell for their assistance and support in processing samples for mass cytometry, and A Lun for his advice and support on analysing mass cytometry data.

## Additional information

### Funding

| Funder | Grant reference number | Author |
|---|---|---|
| Wellcome | 103930 | Gillian M Griffiths |
| Wellcome | 100140 | Gillian M Griffiths |
| Wellcome | 217100 | Gillian M Griffiths |
| Wellcome | 204017/Z/16/Z | Claire Y Ma |
| Cancer Research UK | A17197 | John C Marioni |
| Medical Research Council | MR/P014178/1 | Arianne C Richard |
| Addenbrooke's Charitable Trust, Cambridge University Hospitals | 23/17 A (ii) | Claire Y Ma |
| European Molecular Biology Organization | | John C Marioni |

The funders had no role in study design, data collection and interpretation, or the decision to submit the work for publication.

### Author contributions

Claire Y Ma, Investigation, Resources, Formal analysis, Funding acquisition, Writing - original draft, Writing - review and editing; John C Marioni, Conceptualization, Supervision, Funding acquisition, Methodology, Writing - review and editing; Gillian M Griffiths, Conceptualization, Resources, Supervision, Funding acquisition, Writing - review and editing; Arianne C Richard, Conceptualization, Formal analysis, Supervision, Funding acquisition, Investigation, Methodology, Writing - original draft, Writing - review and editing

### Author ORCIDs

Claire Y Ma ⓘ https://orcid.org/0000-0002-4244-7535
John C Marioni ⓘ https://orcid.org/0000-0001-9092-0852
Gillian M Griffiths ⓘ https://orcid.org/0000-0003-0434-5842
Arianne C Richard ⓘ https://orcid.org/0000-0002-8708-9997

### Ethics

Animal experimentation: Experiments were carried out under Project Licence PPL 70/8590. This research has been regulated under the Animals (Scientific Procedures) Act 1986 Amendment Regulations 2012 following ethical review by the University of Cambridge Animal Welfare and Ethical Review Body (AWERB).

### Decision letter and Author response

Decision letter https://doi.org/10.7554/eLife.53948.sa1
Author response https://doi.org/10.7554/eLife.53948.sa2

## Additional files

### Supplementary files

• Supplementary file 1. Replicate structure. Table details independent experiments, biological replicates and, where applicable, barcoding strategies.

• Supplementary file 2. Key Resources Table.

• Supplementary file 3. Antibodies for mass cytometry.

• Supplementary file 4. Antibody testing and specificity. Antibody specificity tables detail experiments used to test and validate (a) antibodies targeting total proteins and (b-c) signalling proteins. Tables also include relevant references utilising these clones in knockout, knockdown, overexpression or small molecule inhibitor experiments.

• Supplementary file 5. Mass cytometry figure underlying data. Tables provide means and standard deviations of kinetics curves depicted in *Figures 3a* and *4b–c*, and *Figure 3—figure supplement 2a*, as well as summary statistics from hypersphere differential abundance testing depicted in *Figure 4a* and *Figure 4—figure supplement 1*.

• Transparent reporting form

## Data availability

Raw mass cytometry data can be found on the Flow Repository, accession numbers FR-FCM-Z2CX and FR-FCM-Z2CP. Full results of mass cytometry analyses are included as Supplementary File 5. Source data for summary plots of flow cytometry-measured signaling markers in T cells stimulated with peptide-loaded BMDCs (Figure 7a) are included as Figure 7 - Source Data File 1. Analysis code is available at https://github.com/MarioniLab/SignallingMassCytoStimStrength (copy archived at https://github.com/elifesciences-publications/SignallingMassCytoStimStrength).

The following datasets were generated:

| Author(s) | Year | Dataset title | Dataset URL | Database and Identifier |
|---|---|---|---|---|
| Ma CY, Marioni JC, Griffiths GM, Richard AC | 2019 | Ma et al CD8+ T cell signalling panel experiment 2 | http://flowrepository.org/id/FR-FCM-Z2CP | Flow Repository, FR-FCM-Z2CP |
| Ma CY, Marioni JC, Griffiths GM, Richard AC | 2019 | Ma et al CD8+ T cell signalling panel experiment 1 | http://flowrepository.org/id/FR-FCM-Z2CX | Flow Repository, FR-FCM-Z2CX |

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
