## [Decision Letter]

**Acceptance summary:**

You have generated a highly detailed and powerful in vitro analysis of T cell signaling networks triggered by a range of peptide-MHC potencies. The finding that responses to weak ligands activate similar signalling pathways, but with different kinetics, adds to our understanding of how CD8^+^ T cells cope with the wide range of ligand potencies that are encountered physiologically. We also appreciate that you have acknowledged the potential of the Erk1/2 node to encode qualitatively distinct responses to ligand of differing potency, which may also be an important observation for going forward.

**Decision letter after peer review:**

[Editors’ note: the authors submitted for reconsideration following the decision after peer review. What follows is the decision letter after the first round of review.]

Thank you for submitting your work entitled "TCR stimulation strength controls the rate of initiation but not the molecular organization of signalling" for consideration by *eLife*. Your article has been reviewed by three peer reviewers, including Michael L Dustin as the Reviewing Editor and Reviewer #1, and the evaluation has been overseen by a Senior Editor. The following individual involved in review of your submission has agreed to reveal their identity: Christoph Wuelfing (Reviewer #2).

Our decision has been reached after consultation between the reviewers. Based on these discussions and the individual reviews below, we regret to inform you that your work will not be considered further for publication in *eLife*.

Your study illustrates the great potential of multiplexing to analyse outputs of multiple signaling pathways using mass cytometry and you present an interesting hypothesis that ligands of different potencies control kinetics, rather than the nature of signals generated. The weaknesses of the study include lack of information about the outcome of this mode of T cell activation, potential gaps in coverage of relevant signaling pathways that could reveal differences with ligand potency, and circumstantial evidence for the final working model. It was felt that addressing these concerns in 2 months was not possible.

Reviewer #1:

The authors have performed stimulation of OTI T cells in vitro using "self" presentation of peptides introduced into the media for 1-6 hrs followed by mass cytometry or flow cytometry based analysis of signaling proteins, their phospho-forms and selected surface markers and mRNAs. The authors conclude that there are no major differences in the activation of the signaling nodes by the different potency ligands, but only difference in the rate and extent of activation of the same network. This result extends recent results from the same group on single cell transcriptomics that also showed the same gene expression trajectories with different potencies of peptides for OT1 TCR under similar in vitro stimulation conditions. The study is carefully performed generating a data set with two biological replicates from a transgenic mouse model that is expected to have limited variability. Despite the enormous complexity of the dataset, two major clusters A and B and be tracked and appear to occur in the responding cells for all potencies of peptides, but faster and to a great extent for more potent peptides.

1) The schematic in Figure 1 depicts a signaling network down down-stream of T cell interaction with an APC. If I correctly understand the system, the pMHC are being presented by naïve T cells, which express limited CD80 basally, although this may be induced during the 6 hr and this may change differentially with the peptides. The authors should include CD80 and ICAM-1 among the surface markers (perhaps by flow cytometry) to understand other aspects of signal evolution over the first 6 hrs though increased co-stimulation and adhesion. Some details in this figure may un-necessarily add to the complexity- for example- Y192 on Lck doesn't really come into the study and inspection of the data in the paper suggests that while phosphorylation at this site can generate negative feedback, the conditions under which this would happen is not clear and the conditions are unlikely to come into play in the very simple conditions in this paper. I think this figure could be simplified for focus on relevant information, in addition to providing an accurate picture of the experiment that is done- with another T cell as the APC.

2) The authors note that the highest potency peptide induce significant numbers of events in the mass cytometry with 2x DNA. This seems likely to be biologically relevant conjugate formation related to the self-presentation of the pMHC between T cells, which is the only form of antigen presentation in the system. These events are not further analyzed, it would be helpful to have a global analysis of the % of these events for each time point with each peptide as the loss of these events may skew the analysis as the cells engaged in conjugates may be the most highly activated cells, for example, with higher levels of ICAM-1 and CD80 driving stronger conjugate formation. It’s possible that the progression to particularly stable LFA-1 dependent conjugates during T cell activation may also be a relevant phenotype for T cell differentiation as suggested by work from Gerard and colleagues (Gerard A, Khan O, Beemiller P, Oswald E, Hu J, Matloubian M, Krummel MF. Secondary T cell-T cell synaptic interactions drive the differentiation of protective CD8^+^ T cells. Nat Immunol. 2013;14(4):356-63.).

3) The study has limitations in that it focuses on a very simplified antigen presentation process that very much emphasizes the TCR-pMHC interaction in the context of a T-T interaction, which is not the way T cell priming would work. In the CD8 system the major differentiation process relates to whether or how terminal effector and long-lived memory cells are generated. It seems unlikely that T-T presentation would generate any T cell memory so this aspect of terminal differentiation vs generation of cells with stem cell line characteristics, and if this would be impacted by ligand potency, is not addressed. In the bigger picture of CD8 T cell biology some acknowledgement of these issues and how one might bridge these studies to address that issues would be helpful to readers.

4) With only two biological replicates there is a question of statistical significance of the observations. Can the authors comment on the reproducibility of the mass cytometry experiment? Is there a way to attach a statistical likelihood to the possibility that the signaling trajectories are actually different for the different ligands, but this has been missed due to variability of the data and the limited number of replicates? This is an interesting cased where there is a lot of very detailed data from only two observations (plus additional flow cytometry data sets) and there is effectively only one technical replicate as I believe the two biological replicates were run at the same time on the machine with bar coding to distinguish them. Some discussion of how statistical assessment of the conclusion can be made would be very helpful.

Reviewer #2:

Understanding the complexity of T cell signalling requires information-rich data. Mass cytometry with large panels of antibodies against signalling intermediates and their active states is a powerful method to provide such data. Its application to naïve T cell activation is highly welcome. Ma et al. applied an antibody panel determining activity of nine components of the T cell signalling system to the activation of naïve MHC I-restricted T cell receptor transgenic T cells in response to peptides with a range of affinities by having T cells present MHC/peptide complexes to each other at time points between one and six hours. Differential activation of three of the signalling intermediates, Erk, STAT5 and S6, allowed for distinction between different T cell activation conditions such that frequency and/or dynamics of the activation of these signalling intermediates vary. The manuscript powerfully illustrates the potential of mass cytometry to unravel complex mechanisms in T cell signalling. However, a modest scale and a few experimental constraints substantially limit the actual insight gained into T cell activation.

In this manuscript, naïve T cells are activated by presenting peptide to each other. I appreciate that this allows for the unambiguous assignment of mass cytometry signals to the T cells. It also forces T cells to activate with limited availability of costimulation. This may explain why there is virtually no activation of four (fairly proximal) of the nine signalling intermediates assayed, ZAP-70, SLP-76, PLCgamma and Akt. There are no changes in Lck phosphorylation (consistent with Acuto lab data) and CD25 expression. Thus, the nine signalling intermediate panel essentially contracts to four signalling intermediates, Erk, STAT5, S6 and IκB. This removes a key strength of mass cytometry from this study, the ability to determine relations between significant changes across numerous events. Some signalling intermediates are preferentially expressed in T cells, e.g. LAT and SLP-76. They, therefore, could be unambiguously detected in T cells activated more efficiently by professional APCs such as dendritic cells. In addition, as mass cytometry is a single cell approach, T cells could be dissociated from activating APCs before processing for mass cytometry and T cell-specific events identified with antibodies against T cell surface markers, such as the TCR or CD8. Even though a complete study using professional APCs such as DCs would be well beyond the scope of any revision, any insight from pilot experiments whether such APCs could trigger more effective signalling that can be unambiguously be assigned to the T cells would be of great use for other investigators planning to use mass cytometry to investigate T cell signalling.

The authors decided to supply exogenous IL-2. As IL-2 provides auto/paracrine amplification of T cell activation this likely has led to less variable data sets. However, increased distinction between T cell activation conditions provided by such amplification could have also provided additional physiological insight into mechanisms of T cell signal discrimination. Again, some initial insight from pilot studies that would extend the determination of the role of exogenous IL-2 from STAT5 and SP6 to other signalling intermediates could be of great value to other investigators.

One of the key challenges in T cell signalling is to connect signalling on the time scale of minutes to hours to cell fate on the time scale of days. While there is plenty of data in general on the response of naïve OT-I T cells to the peptide variants used , e.g. in the induction of proliferation, cytokine secretion and cytolytic capability, it would be good if such data directly related to the specific naïve T cell activation conditions used here, i.e. peptide presentation between naïve T cells. This is of particular interest as signalling differences at the last, 6-hour, time point in response to the different peptides are surprisingly small (as beautifully summarised in Figure 4—figure supplement 2 and also apparent in Figure 6B). This raises important questions. With signalling differences at 6 hours that small, what are the consequences for presumably substantially divergent later activation events? If later differences are more closely related to signalling at 2 to 4 hours, how are such differences “remembered”? Do we need to find additional signalling events to effectively link signalling on the time scale of hours to consequences on the time scale of days? Thus, knowing standard T cell activation outcomes in response to the different peptides under the specific experimental T cell activation conditions used in this study would be very useful.

In summary, the manuscript powerfully illustrates how much and interesting data can in principle be generated by applying mass cytometry approaches to T cell activation. In its current form, however, physiological insight remains limited by the small effective size of the antibody panel (only three discriminatory markers), the choice of T cell activation without APC and upon addition of exogenous IL-2 and the lack of direct relation to long-term outcomes of T cell signalling under the chosen experimental conditions. An expansion of the antibody panel is certainly beyond the scope of any reasonable revision. However, pilot experiments to illustrate consequences of the experimental choices together with a determination of long-term T cell activation consequences and an acknowledgement of limitations in physiological insight caused by the small effective antibody panel could still make this an influential manuscript for future work on unravelling T cell signalling complexity with mass cytometry.

Reviewer #3:

Ma at al. used the CyTof technology to determine in parallel and with single cell resolution several molecules that are involved in TCR signal transduction. They used this approach with the intention to gain new insights into how different qualities of TCR ligation (high versus low affinity) translate into different T cell responses.

This is in principle a highly relevant and in key parts still barely understood matter. However, I have several issues with the manuscript of which the key points are i) lack of novelty and ii) that only circumstantial evidence is presented.

I see only incremental new insights that in my opinion do not move the field forward or provide real answers. Moreover, the authors want to put the message forward that mainly kinetic difference and not the signals quality determine the outcome of high versus low affinity signaling. While this is an interesting hypothesis, the authors present only circumstantial evidence and do not establish a causal relationship.

Other issues involve that the authors mainly use a fixed concentration of peptide which is with 1uM quite high, while titration experiments are rare. Moreover, would the same signaling pattern be observed when the cells are exposed to the same ligands in vivo?

[Editors’ note: further revisions were suggested prior to acceptance, as described below.]

Thank you for submitting your article "Stimulation strength controls the rate of initiation but not the molecular organization of TCR-induced signalling" for consideration by *eLife*. Your article has been reviewed by two peer reviewers, and the evaluation has been overseen by a Reviewing Editor and Aleksandra Walczak as the Senior Editor The following individuals involved in review of your submission have agreed to reveal their identity: Christoph Wuelfing (Reviewer #2).

The reviewers have discussed the reviews with one another and the Reviewing Editor has drafted this decision to help you prepare a revised submission.

We would like to draw your attention to changes in our revision policy that we have made in response to COVID-19 (https://elifesciences.org/articles/57162). Specifically, we are asking editors to accept without delay manuscripts, like yours, that they judge can stand as *eLife* papers without additional data, even if they feel that they would make the manuscript stronger. Thus, the revisions requested below only address clarity and presentation.

Your paper provides extremely detailed study of the kinetics of different activation responses in individual CD8 T cells triggered by TCR ligands with a well-defined hierarchy of affinities. You have also been very responsive to the comments of the reviewers in the first round, clarifying the statistical strength of their experimental design and adding key experiments in response to reviewer comments.

The reviewers and editors nonetheless suggest essential revision to address the potential of pErk1/2 to encode information about the different ligands that may not be fully read out in your otherwise thorough analysis. One effect that stands out in the data appears to be an affinity-related all-or- none involvement of the ERK pathway. Activation or lack of activation of pERK distinguishes two clusters of otherwise-coordinate responses, and while this pERK activation is frequent and early in cells activated by high-affinity ligands, it is hardly detectable at all in cells activated with low-affinity ligands, even at later time-points when all the other pathways have become activated. The fact that all the other pathways tested show kinetic but not qualitative differences in response to high and low affinity ligands makes the distinctive behavior of ERK very interesting.

You state that: "Given that we found full S6 phosphorylation requires MEK signalling, we hypothesize that transient ERK activation events (Ryu et al., 2018) occurred in many cells outside of the snapshots of our measurements." You assume that if you used a "real-time" reporter that G4 would cause pErk1/2 to "blink" at a frequency sufficient to account for the Mek contribution of pS6, but very rarely captured in the snapshot? It might be helpful to be more explicit about this or even make this prediction if it’s how you account for the result. It seems that what we know about pErk activation could fit with this and in addition to Ryo et al. it might make sense to cite Huang WYC, Alvarez S, Kondo Y, Lee YK, Chung JK, Lam HYM, Biswas KH, Kuriyan J, Groves JT. A molecular assembly phase transition and kinetic proofreading modulate Ras activation by SOS. Science. 2019;363(6431):1098-103. Epub 2019/03/09. doi: 10.1126/science.aau5721. PubMed PMID: 30846600; PMCID: PMC6563836 as it makes a case for how Erk activation down stream of Sos could have this property. Is it possible that even though this blinking is sufficient for pS6, that perhaps other TF like Elk might decode this differently and allow qualitatively different responses to these ligands? While the Erk1/2 signal may be showing such blinking, this is not demonstrated here and the pS6 phosphorylation could be driven by other signals than Erk1/2. Figure 5 is actually misleading in the absence of direct evidence for the pErk in the G4 cells. This requires a more critical discussion as it’s not clear you have proven that the cell will ignore the information encoded in the pERK1/2 node even if your untested working model is correct.

---

## [Author Response]

[Editors’ note: The authors appealed the original decision. What follows is the authors’ response to the first round of review.]

Reviewer #1:1) The schematic in Figure 1 depicts a signaling network down down-stream of T cell interaction with an APC. If I correctly understand the system, the pMHC are being presented by naïve T cells, which express limited CD80 basally, although this may be induced during the 6 hr and this may change differentially with the peptides. The authors should include CD80 and ICAM-1 among the surface markers (perhaps by flow cytometry) to understand other aspects of signal evolution over the first 6 hrs though increased co-stimulation and adhesion.

We have now measured CD80 and ICAM-1 on the surface of naïve T cells after 6 hours of stimulation with strong, medium, weak or null ligands (new Figure 7—figure supplement 1A). These results demonstrate that ICAM-1 expression is unchanged between stimuli and CD80 expression is absent. As the ICAM-1 ligand LFA-1 is expressed by naïve T cells, ICAM-1-LFA-1 interactions may therefore play a role in the stimulation system we used, but CD80-CD28 co-stimulation is unlikely to. For comparison with professional antigen-presenting cells, we examined the same molecules on the surface of BMDCs (new Figure 7—figure supplement 1B), which were found to express high levels of both CD80 and ICAM-1. These results and their implications are now described in the text as follows:

“The interaction of adhesion molecules LFA-1 and ICAM-1 assists the formation of a stable immunological synapse, augments TCR-induced signalling, and continues to promote differentiation even after initial activation (Gérard et al., 2013; Verma and Kelleher, 2017). […] In contrast, mature bone marrow-derived dendritic cells (BMDCs) expressed high levels of CD80, along with additional costimulatory molecules (Figure 7—figure supplement 1B).”

Some details in this figure may un-necessarily add to the complexity- for example- Y192 on Lck doesn't really come into the study and inspection of the data in the paper suggests that while phosphorylation at this site can generate negative feedback, the conditions under which this would happen is not clear and the conditions are unlikely to come into play in the very simple conditions in this paper. I think this figure could be simplified for focus on relevant information, in addition to providing an accurate picture of the experiment that is done- with another T cell as the APC.

We have modified Figure 1 to depict only the signalling pathways we measured and more directly model the systems we investigated.

2) The authors note that the highest potency peptide induce significant numbers of events in the mass cytometry with 2x DNA. This seems likely to be biologically relevant conjugate formation related to the self-presentation of the pMHC between T cells, which is the only form of antigen presentation in the system. These events are not further analyzed, it would be helpful to have a global analysis of the % of these events for each time point with each peptide as the loss of these events may skew the analysis as the cells engaged in conjugates may be the most highly activated cells, for example, with higher levels of ICAM-1 and CD80 driving stronger conjugate formation. It’s possible that the progression to particularly stable LFA-1 dependent conjugates during T cell activation may also be a relevant phenotype for T cell differentiation as suggested by work from Gerard and colleagues (Gerard A, Khan O, Beemiller P, Oswald E, Hu J, Matloubian M, Krummel MF. Secondary T cell-T cell synaptic interactions drive the differentiation of protective CD8^+^ T cells. Nat Immunol. 2013;14(4):356-63.).

We have added new Figure 2—figure supplement 2B to show the percentage of doublet events across conditions.

We thank the reviewer for pointing out the findings of Gerard et al. The critical differentiation period Gerard et al. identified occurs roughly 24 hours after stimulation when T-T synapses allow cytokine exchange. While highly relevant for subsequent differentiation, this process is likely too late to examine in the short stimulation time courses we ran. Instead of we have added text to highlight the importance of LFA-1-ICAM-1 interactions, including a citation of this role identified by Gerard et al. (as described in response to point 1).

3) The study has limitations in that it focuses on a very simplified antigen presentation process that very much emphasizes the TCR-pMHC interaction in the context of a T-T interaction, which is not the way T cell priming would work. In the CD8 system the major differentiation process relates to whether or how terminal effector and long-lived memory cells are generated. It seems unlikely that T-T presentation would generate any T cell memory so this aspect of terminal differentiation vs generation of cells with stem cell line characteristics, and if this would be impacted by ligand potency, is not addressed. In the bigger picture of CD8 T cell biology some acknowledgement of these issues and how one might bridge these studies to address that issues would be helpful to readers.

We agree that our in vitro system is skewed toward effector differentiation but believe that there are advantages to using such a reductionist system to investigate the impact of signal strength. By minimizing inputs from co-stimulation, we were able to ask how modulating *only* the strength of the TCR-pMHC interaction affects signalling pathways. We have clarified the inherent effector bias of this culture system as follows:

“We used a minimal, controlled system of peptide addition, allowing T cells to present antigens to each other. We also added exogenous IL2 to mitigate effects of potency-dependent differences in paracrine IL2 (Au-Yeung et al., 2017; Denton et al., 2011; Marchingo et al., 2014; Tan et al., 2017; Verdeil, Puthier, Nguyen, Schmitt-Verhulst, and Auphan-Anezin, 2006; Guillaume Voisinne et al., 2015) and provide all cells with an effector-promoting environment (Pipkin et al., 2010; Verdeil et al., 2006). This system was chosen in order to examine the cell-intrinsic effects of TCR stimulation strength on signalling pathways.”

We have also added new data examining naïve T cells stimulated by peptide-loaded BMDCs (new Figure 7, as described in response to reviewer 2 point 1). Finally, we have added text about the benefits and limitations of the T:T presentation system, as well as how our in vitro results might relate to in vivo T cell differentiation:

Results: “In contrast to this reductionist system, many additional factors impact T cell activation in vivo. Most fundamentally, naïve T cells are activated in the lymph node by professional antigen-presenting cells (APCs), such as dendritic cells, instead of other T cells. These APCs express costimulatory ligands in addition to peptide-MHC complexes, which can further tune naïve T cell responses (L. Chen and Flies, 2013; Hubo et al., 2013).”

Discussion: “… this in vitro system is nevertheless still far-removed from in vivo T cell activation, where the microenvironment is increasingly complex. […] By delineating the impact of stimulation strength in low-complexity systems, these data can form the basis for interpretation of future studies where additional variables may be explored.”

4) With only two biological replicates there is a question of statistical significance of the observations. Can the authors comment on the reproducibility of the mass cytometry experiment? Is there a way to attach a statistical likelihood to the possibility that the signaling trajectories are actually different for the different ligands, but this has been missed due to variability of the data and the limited number of replicates? This is an interesting cased where there is a lot of very detailed data from only two observations (plus additional flow cytometry data sets) and there is effectively only one technical replicate as I believe the two biological replicates were run at the same time on the machine with bar coding to distinguish them. Some discussion of how statistical assessment of the conclusion can be made would be very helpful.

We apologise for any confusion about the number of replicates included in mass cytometry experiments and lack of clarity about statistical analyses. Six biological replicates were examined by mass cytometry, although not all had data for all time points. We have added tables delineating the replicate structure and statistics for replicate measurements of individual signalling molecules and multidimensional phenotypes (new Supplementary files 1 and 5).

We have also added an expanded description of the cydar method for mass cytometry differential abundance analysis:

“To test for differential abundance of cells with any combinatorial phenotype, agnostic to cellular density or clustering patterns, we employed cytometry differential abundance testing in R (cydar, (A. T. L. Lun et al., 2017)). This method takes advantage of the consistent staining achieved with sample barcoding, along with the count-based nature of single cell data, to find regions of the high-dimensional marker space occupied significantly more or less frequently by cells from a particular condition. This is achieved by filling the marker space with hyperspheres, comparing cellular abundances within each hypersphere across conditions, and controlling the false discovery rate across the marker space.”

We used cydar to test for differentially abundant populations between any stimulation condition and unstimulated controls. Due to cydar’s cluster-free approach, drift in the mass cytometry detector between tubes can falsely appear as differential abundance of subtly different cellular phenotypes. For this reason, only experiments that are directly multiplexed or include a subset of shared samples that can be used for normalization are compatible with this method (Lun ATL et al., 2017). Therefore, cydar statistical tests were run on a mass cytometry experiment comprised of 2 biological replicates of the full time-course with a strategically multiplexed design that maximized the number of samples that could be included within the 20 mass cytometry barcodes available from Fluidigm (as detailed in Supplementary file 1). All reported differentially abundant phenotypes met statistical significance using a false discovery rate threshold of 5 %. Using phenotypes identified by this differential abundance analysis (Figure 4A), we then specifically gated this mass cytometry experiment (2 biological replicates) and a second experiment (4 biological replicates at overlapping time points) to generate the kinetics plots of the various phenotypes observed and confirm their reproducibility across more replicates (Figure 4B-C, statistics provided in new Supplementary file 5). Additional replication of certain markers was provided by flow cytometry analyses throughout the manuscript.

We appreciate the reviewer’s request for a statistical assessment of the likelihood that signalling trajectories are shared across stimuli. We have now plotted the inferred trajectories of activation events described in Figure 5 (pS6, pERK, pSTAT5, CD44) for cells stimulated with each ligand (new Figure 5—figure supplement 1A-C). We also added an explicit test of the probability that the order of activation events occurring after stimulation with each ligand would be as similar as observed if all events occurred at random. These analyses are described in the text (Materials and methods and Results) as follows:

Materials and methods: “For trajectory analysis, each biological replicate was analysed separately. A logicle transformation (default parameters except w=0.1) was applied to raw intensity data. […] It was not possible to order events in the remaining biological replicates from experiment 1 that included only 1 stimulated timepoint.”

Results: “To formally test this order of activation events, we constructed activation trajectories of cells under each stimulation condition across all time points, based on their expression of pS6 (Figure 5—figure supplement 1A). […] The signalling molecules pAKT, pLCK and IκBα were less dynamic along the trajectory, precluding precise determination of their order of activation particularly in weakly stimulated cells, but visualizing their changes along the trajectory further suggested shared patterns between stimuli (Figure 5—figure supplement 1D).”

Reviewer #2:[…] The manuscript powerfully illustrates the potential of mass cytometry to unravel complex mechanisms in T cell signalling. However, a modest scale and a few experimental constraints substantially limit the actual insight gained into T cell activation.In this manuscript, naïve T cells are activated by presenting peptide to each other. I appreciate that this allows for the unambiguous assignment of mass cytometry signals to the T cells. It also forces T cells to activate with limited availability of costimulation. This may explain why there is virtually no activation of four (fairly proximal) of the nine signalling intermediates assayed, ZAP-70, SLP-76, PLCgamma and Akt. There are no changes in Lck phosphorylation (consistent with Acuto lab data) and CD25 expression. Thus the nine signalling intermediate panel essentially contracts to four signalling intermediates, Erk, STAT5, S6 and IκB. This removes a key strength of mass cytometry from this study, the ability to determine relations between significant changes across numerous events. Some signalling intermediates are preferentially expressed in T cells, e.g. LAT and SLP-76. They, therefore, could be unambiguously detected in T cells activated more efficiently by professional APCs such as dendritic cells. In addition, as mass cytometry is a single cell approach, T cells could be dissociated from activating APCs before processing for mass cytometry and T cell-specific events identified with antibodies against T cell surface markers, such as the TCR or CD8. Even though a complete study using professional APCs such as DCs would be well beyond the scope of any revision, any insight from pilot experiments whether such APCs could trigger more effective signalling that can be unambiguously be assigned to the T cells would be of great use for other investigators planning to use mass cytometry to investigate T cell signalling.

We agree that examination of signalling after stimulation with professional antigen presenting cells is of great interest. We have now activated naïve T cells with peptide-pulsed BMDCs and examined their signalling at selected proximal and distal signalling nodes (pZAP70, pSLP76, pERK1/2, pS6, pSTAT5), as well as CD44 expression, by flow cytometry (new Figure 7). These data showed that pSLP76, pERK1/2 and CD44 responded in potency-dependent manners similar to T-cell-T-cell antigen presentation, while pS6 was upregulated with high, medium and low potency stimuli in a potency-independent manner, and pSTAT5 was activated simply by co-culture with BMDCs. These results are described and discussed in the text (Results and Discussion) as follows:

Results: “To test how signalling responses to ligands of different strengths might be impacted by the additional signalling conferred by professional APCs, we stimulated naïve T cells with mature BMDCs (Figure 7—figure supplement 1B-C) loaded with peptides of various potencies. […] These results indicate that the rate-based mechanism we observed in the T:T stimulation system is further tuned at particular signalling nodes by more complex antigen presentation.”

Discussion: “Under stimulation with peptide-loaded BMDCs, ligand potency determined the percentages of T cells undergoing certain activation events (pERK1/2, pSLP76 and CD44), similar to observations in our reductionist stimulation system. […] Further exploration of how individual costimulatory ligands impact the coordination and initiation rate of the TCR-induced signalling programme will be important for dissecting these additional inputs.”

The authors decided to supply exogenous IL-2. As IL-2 provides auto/paracrine amplification of T cell activation this likely has led to less variable data sets. However, increased distinction between T cell activation conditions provided by such amplification could have also provided additional physiological insight into mechanisms of T cell signal discrimination. Again, some initial insight from pilot studies that would extend the determination of the role of exogenous IL-2 from STAT5 and SP6 to other signalling intermediates could be of great value to other investigators.

We have now examined the impact of IL2 on pAKT, pERK and IκBα signalling (newly expanded Figure 3—figure supplement 2B) and have found no impact of IL2 on signalling at pERK or IκBα, and a subtle effect on pAKT in our stimulation conditions. These results are described in the text as follows:

“The percentages of cells degrading IκBα or phosphorylating S6 or ERK1/2 were not impacted by the presence of exogenous IL2. The percentages of cells phosphorylating pAKT[S473] were subtly increased by IL2 particularly under stimulation with low potency ligands (Figure 3—figure supplement 2B). This may reflect the mechanism proposed by Ross et al. whereby JAK signalling induced by IL2 ultimately stimulates mTORC2 phosphorylation of AKT[S473] (Ross et al., 2016).”

One of the key challenges in T cell signalling is to connect signalling on the time scale of minutes to hours to cell fate on the time scale of days. While there is plenty of data in general on the response of naïve OT-I T cells to the peptide variants used , e.g. in the induction of proliferation, cytokine secretion and cytolytic capability, it would be good if such data directly related to the specific naïve T cell activation conditions used here, i.e. peptide presentation between naïve T cells. This is of particular interest as signalling differences at the last, 6-hour, time point in response to the different peptides are surprisingly small (as beautifully summarised in Figure 4—figure supplement 2 and also apparent in Figure 6B). This raises important questions. With signalling differences at 6 hours that small, what are the consequences for presumably substantially divergent later activation events? If later differences are more closely related to signalling at 2 to 4 hours, how are such differences “remembered”? Do we need to find additional signalling events to effectively link signalling on the time scale of hours to consequences on the time scale of days? Thus, knowing standard T cell activation outcomes in response to the different peptides under the specific experimental T cell activation conditions used in this study would be very useful.

We thank the reviewer for highlighting this point of interest. The in vitro T cell activation system we use in the presence of exogenous IL-2 promotes effector differentiation, as described in our response to reviewer 1, point 3. We previously conducted a study using this activation system in which we profiled mRNA expression after 6 hours, and CD44 protein expression and proliferation after 2 days (Richard et al., 2018). These data demonstrated that, regardless of stimulation strength, T cells followed the same main transcriptional activation pathways and were capable of proliferation and CD44 expression. Instead, stimulation strength altered the probability that a given cell would initiate transcriptional activation. We have expanded the text to describe these outcomes as follows:

“Using this reductionist stimulation system, we previously found that stimulation strength determined the rate with which naïve T cells initiated transcriptional activation but that cells activated by all ovalbumin-derived ligands were proliferating and expressing the effector molecule CD44 by two days (Richard et al., 2018).”

In summary, the manuscript powerfully illustrates how much and interesting data can in principle be generated by applying mass cytometry approaches to T cell activation. In its current form, however, physiological insight remains limited by the small effective size of the antibody panel (only three discriminatory markers), the choice of T cell activation without APC and upon addition of exogenous IL-2 and the lack of direct relation to long-term outcomes of T cell signalling under the chosen experimental conditions. An expansion of the antibody panel is certainly beyond the scope of any reasonable revision. However, pilot experiments to illustrate consequences of the experimental choices together with a determination of long-term T cell activation consequences and an acknowledgement of limitations in physiological insight caused by the small effective antibody panel could still make this an influential manuscript for future work on unravelling T cell signalling complexity with mass cytometry.

We appreciate that the reviewer sees the potential of mass cytometry to address questions of T cell signalling and fully acknowledge that our study is limited in the number of dynamic markers tested. We believe that the use of a minimal in vitro system adding pure peptide in the presence of IL2 allowed us to examine the impact of stimulation strength on *cell-intrinsic* signalling processes without contributions from costimulation or autocrine/paracrine IL2 feedback. Understanding these effects can then facilitate more nuanced interpretation of increasingly complex systems. In addition to adding experiments that begin to extend these findings (including additional signalling events profiled in the absence of IL2 and after stimulation with peptide-pulsed BMDCs), we have also added discussion of the rationale and limitations of our stimulation system and how results may differ in more physiological settings as described in response to reviewer 1 point 3.

Reviewer #3:This is in principle a highly relevant and in key parts still barely understood matter. However, I have several issues with the manuscript of which the key points are i) lack of novelty and ii) that only circumstantial evidence is presented.I see only incremental new insights that in my opinion do not move the field forward or provide real answers. Moreover, the authors want to put the message forward that mainly kinetic difference and not the signals quality determine the outcome of high versus low affinity signaling. While this is an interesting hypothesis, the authors present only circumstantial evidence and do not establish a causal relationship.

While we agree that the insights we were able to gather from our study were limited by the number of detectable dynamic markers, we believe that this information is nonetheless valuable to share across the field so that future efforts can build upon our experimental set-up and results as described in response to reviewer 1 point 3.

In light of the limited number of dynamic signalling molecules we profiled in the current study and to address the possibility that other unmeasured signalling events do not respond to different strengths of stimulation with altered activation rates, we have also added appropriate caveats to our conclusions in the text (Abstract, Introduction and Discussion):

Abstract: “Using mass cytometry to simultaneously measure multiple signalling pathways, we found a programme of distal signalling events that is shared, regardless of the strength of TCR stimulation.”

Introduction: “Our data suggest that the coordination of the TCR-induced signalling pathways that we tested does not vary with stimulation strength.”

Discussion: “In this study, we measured 22 markers of protein expression and active signalling. While other unmeasured signalling mediators may respond to altered stimulation strength in a different manner, our data demonstrate a strict choreography of the distal signalling processes that we examined.”

With respect to how the observed kinetic differences in signalling might relate to the ultimate outcomes of high versus low potency signalling, particularly in a physiological setting, we have modified the text as described in response to reviewer 1 point 3 and reviewer 2 points 3 and 4.

Other issues involve that the authors mainly use a fixed concentration of peptide which is with 1uM quite high, while titration experiments are rare. Moreover, would the same signaling pattern be observed when the cells are exposed to the same ligands in vivo?

The goal of our study was to examine the impact of stimulation strength. We therefore sought to saturate the antigen presentation and response machinery to limit the influence of concentration-dependent differences. We previously found that a 1 μM concentration of these peptides saturated changes in surface protein expression after 4 hours of stimulation (Richard et al., 2018), and in the current manuscript we demonstrated the same for pERK and pS6 signalling at 2 and 4 hours (Figure 2—figure supplement 1).

Regarding how our in vitro system would compare to in vivo responses, please see the response to reviewer 1 point 3 and reviewer 2 point 4.

[Editors’ note: what follows is the authors’ response to the second round of review.]

The reviewers and editors nonetheless suggest essential revision to address the potential of pErk1/2 to encode information about the different ligands that may not be fully read out in your otherwise thorough analysis. One effect that stands out in the data appears to be an affinity-related all-or- none involvement of the ERK pathway. Activation or lack of activation of pERK distinguishes two clusters of otherwise-coordinate responses, and while this pERK activation is frequent and early in cells activated by high-affinity ligands, it is hardly detectable at all in cells activated with low-affinity ligands, even at later time-points when all the other pathways have become activated. The fact that all the other pathways tested show kinetic but not qualitative differences in response to high and low affinity ligands makes the distinctive behavior of ERK very interesting.You state that: "Given that we found full S6 phosphorylation requires MEK signalling, we hypothesize that transient ERK activation events (Ryu et al., 2018) occurred in many cells outside of the snapshots of our measurements." You assume that if you used a "real-time" reporter that G4 would cause pErk1/2 to "blink" at a frequency sufficient to account for the Mek contribution of pS6, but very rarely captured in the snapshot? It might be helpful to be more explicit about this or even make this prediction if it’s how you account for the result. It seems that what we know about pErk activation could fit with this and in addition to Ryo et al. it might make sense to cite Huang WYC, Alvarez S, Kondo Y, Lee YK, Chung JK, Lam HYM, Biswas KH, Kuriyan J, Groves JT. A molecular assembly phase transition and kinetic proofreading modulate Ras activation by SOS. Science. 2019;363(6431):1098-103. Epub 2019/03/09. doi: 10.1126/science.aau5721. PubMed PMID: 30846600; PMCID: PMC6563836 as it makes a case for how Erk activation down stream of Sos could have this property.

We agree with the reviewer’s interpretation of our hypothesized mechanism and thank the reviewer for calling our attention to the relevant paper by Huang et al. We have revised the Discussion as follows to clarify our hypothesis and incorporate additional references:

“Given that we found full S6 phosphorylation after strong stimulation requires MEK signalling, we hypothesize that this pathway is active in all stimulation conditions but that ERK activation events occur with reduced frequency or duration with weak stimuli and thus many were missed in our snapshot measurements. […] Such an effect on digital ERK activation may be modulated by multi-step activation of the upstream mediator SOS dependent on its dwell-time after activation-induced recruitment to the plasma membrane (Huang et al., 2019).”

Is it possible that even though this blinking is sufficient for pS6, that perhaps other TF like Elk might decode this differently and allow qualitatively different responses to these ligands?

This is an important point, and we have added the following to the Discussion:

“Interestingly, using a light-inducible ERK activation system in epithelial cells, Aoki et al. demonstrated divergent transcriptional effects of sustained versus transient ERK activation (Aoki et al., 2013). It therefore remains possible that different ERK targets in T cells, such as translational machinery, microtubule remodelling, and transcription factors (e.g. ELK1, SAP1, SAP2) (Navarro and Cantrell, 2014) are differentially affected by stimulation strength, warranting further investigation of additional downstream components.”

While the Erk1/2 signal may be showing such blinking, this is not demonstrated here and the pS6 phosphorylation could be driven by other signals than Erk1/2.

Although we found in Figure 3 that under N4 stimulation, the percentage of cells phosphorylating S6 [S235/S236] was reduced MEK162 treatment, we agree that it remains possible that this mechanism is not the same under G4 stimulation. We have therefore added a call for future investigation of this prediction as described in point (A).

Figure 5 is actually misleading in the absence of direct evidence for the pErk in the G4 cells.

Figure 5 reflects the kinetics we observed in Figures 3 and 4 where the percentages of cells exhibiting the phenotypes (ie the direct evidence) are provided. We have expanded the legend to Figure 5 to clarify this point:

“Cartoon depicts the kinetics of the 4 main signalling phenotypes in cells stimulated with ligands of varying potencies (N4, T4, G4) over time from data in Figures 3A and 4A-B (note the transient pERK^+^ populations, even with G4). Black outlines indicate CD44+ populations.”

This requires a more critical discussion as it’s not clear you have proven that the cell will ignore the information encoded in the pERK1/2 node even if your untested working model is correct.

The Discussion has been modified as described in the response above.